# Compound Warm-Dry and Cold-Wet Events Over the Mediterranean

Paolo De Luca[1,2], Gabriele Messori[2,3], Davide Faranda[4,5], Philip J. Ward[1], and Dim Coumou[1,6]

[1]Department of Water and Climate Risk, Vrije Universiteit Amsterdam, Amsterdam, the Netherlands
[2]Department of Earth Sciences, Uppsala University and Centre of Natural Hazards and Disaster Science (CNDS), Uppsala, Sweden
[3]Department of Meteorology, Stockholm University and Bolin Centre for Climate Research, Stockholm, Sweden
[4]Laboratoire des Sciences du Climat et de l'Environnement, LSCE/IPSL, CEA-CNRS-UVSQ, Université Paris-Saclay, Gif-sur-Yvette, France
[5]London Mathematical Laboratory, London, UK
[6]Royal Netherlands Meteorological Institute (KNMI), De Bilt, the Netherlands

**Correspondence:** Paolo De Luca (p.deluca@vu.nl)

**Abstract.** The Mediterranean (MED) basin is a climate change hot-spot that has seen drying and a pronounced increase in heatwaves over the last century. At the same time, it is experiencing increased heavy precipitation during wintertime cold spells. Understanding and quantifying the risks from compound events over the MED is paramount for present and future disaster risk reduction measures. Here, we apply a novel method to study compound events based on dynamical systems theory and analyse compound temperature and precipitation events over the MED from 1979 to 2018. The dynamical systems analysis quantifies the strength of the coupling between different atmospheric variables over the MED. Further, we consider compound warm-dry anomalies in summer and cold-wet anomalies in winter. Our results show that these warm-dry and cold-wet compound days are associated with large values of the temperature-precipitation coupling parameter of the dynamical systems analysis. This indicates that there is a strong interaction between temperature and precipitation during compound events. In winter, we find no significant trend in the coupling between temperature and precipitation. However in summer, we find a significant upward trend which is likely driven by a stronger coupling during warm and dry days. Thermodynamic processes associated with long-term MED warming can best explain the trend, which intensifies compound warm-dry events.

## 1 Introduction

The Mediterranean (MED) basin is considered a climate change hot-spot (Giorgi, 2006) and has seen winter drying as well as a pronounced increase in summer heatwaves over recent decades (e.g., Mariotti, 2010; Hoerling et al., 2012; Shohami et al., 2011; Nykjaer, 2009). Summer heatwave trends observed over the historical period are mainly driven by thermodynamic changes, such as increasing temperatures, that exacerbate soil drying and daily maximum temperatures. Drying trends during winter are associated with atmospheric circulation changes (i.e. northward shift and intensification of the storm track), likely triggered by increased greenhouse gas and aerosol forcing (Hoerling et al., 2012). However, wintertime heavy precipitation,

often in the form of snowfall, has not decreased as rapidly as one may expect as a consequence of global warming (Faranda, 2019).

Many studies have investigated climate change projections over the MED under high greenhouse gases emission scenarios, providing strong evidence for a continuation of the trends witnessed in the historical period, and much warmer and drier conditions by the end of the $21^{st}$ century (Zappa et al., 2015; Mariotti et al., 2015; Scoccimarro et al., 2016; Hochman et al.,
2018; Samuels et al., 2018; Seager et al., 2014; Barcikowska et al., 2020; Goubanova and Li, 2007; Giorgi and Lionello, 2008; Giannakopoulos et al., 2009; Beniston et al., 2007). Such climatic changes imply more severe and frequent summer heatwaves and droughts (Fischer and Schär, 2010; Giorgi and Lionello, 2008; Beniston et al., 2007; Giannakopoulos et al., 2009), but also an increase in heavy precipitation events notwithstanding the decline in total precipitation (Scoccimarro et al., 2016; Samuels et al., 2018; Goubanova and Li, 2007; Giannakopoulos et al., 2009; Tramblay and Somot, 2018). Changes, such as a reduction
of cold spell intensity, are also expected during winter. For example, Hochman et al. (2020) showed that Cyprus Lows – synoptic low-pressure systems that develop over the Eastern MED and can drive cold spells and heavy precipitation over the Levant – are projected to decrease in frequency and rain-bearing capacity in the future. Changes in atmospheric dynamics, such as an amplified "monsoon-desert mechanism" in summer (Rodwell and Hoskins, 1996; Cherchi et al., 2016; Kim et al., 2019; Wang et al., 2012) or a poleward shift of the tropical belt in winter (Hu and Fu, 2007; Seidel et al., 2008; Peleg et al., 2015;
Totz et al., 2018), may play a significant role in enhancing the drying of the MED in future climates.

In recent years, it has become increasingly clear that hydro-meteorological impacts often result from the compounding nature of several variables and\or events, even if they are not extreme when analysed independently (e.g., Moftakhari et al., 2017; Zscheischler et al., 2020). For natural hazards it is thus important to consider compound, or multi-variate, events (e.g., Zscheischler et al., 2020, 2018; De Luca et al., 2017; De Luca et al., 2020; Couasnon et al., 2020; Ward et al., 2018), as well
as cascading events (e.g., de Ruiter et al., 2020). Such compound events can lead to socio-economic damages exceeding those expected if the individual hazards were to occur separately (e.g., de Ruiter et al., 2020; Barriopedro et al., 2011). The MED region is highly vulnerable to compound heat-related events, such as the co-occurrence of heatwaves and droughts (Manning et al., 2019; Zampieri et al., 2017; Li et al., 2009). Wintertime cold-wet events, especially when associated with snowfall, may also result in costly regional impacts (e.g., Hochman et al., 2019; Bisci et al., 2012). Summer heatwaves and droughts may
lead to premature deaths and wildfires, as occurred during the 2003 and 2010 European heatwaves (Shaposhnikov et al., 2014; Bosch, 2003). On the other hand, cold-wet events during winter may cause road-network disruptions (Seeherman and Liu, 2015).

Here, we specifically seek to characterise precipitation-temperature compound events over the MED in terms of the coupling between precipitation and temperature fields. This allows us to relate long-term changes in compound events to their underlying
physical drivers. We focus on compound warm-dry and cold-wet events during summer (June-July-August, JJA) and winter (December-January-February, DJF), respectively. We apply a method based on dynamical systems theory that reflects the dynamical evolution of the atmosphere and is well-suited to diagnosing changes in atmospheric properties (Faranda et al.,

2019). Our approach considers the analysed variables in terms of their evolution in phase-space, and quantifies the strength of their coupling along with a measure of their persistence (Faranda et al., 2020; De Luca et al., 2020). The article is structured as follows: Section 2 describes the methods, data and statistical tests. Sections 3-4 present the results. Specifically, Section 3 focuses on the strength of the dynamical coupling, chiefly during JJA. Section 4 investigates the large-scale patterns of sea-level pressure (SLP), temperature and precipitation observed during the days when the dynamical coupling is high in both JJA and DJF, and relates these to the compound warm-dry and cold-wet events. Finally, Section 5 summarises and discusses our main findings, and outlines future research opportunities.

## 2 Methods and data

### 2.1 Dynamical systems metrics

In this study, we use a dynamical systems approach to compute two metrics: $\theta^{-1}$ and $\alpha$. The metric $\theta^{-1}$, which we term *persistence*, is very intuitively a measure of the average residence time of the system around a state of interest. Hence, the higher the value of $\theta^{-1}$, the more likely it is that the preceding and future states of the system will resemble the current state over relatively long timescales (Faranda et al., 2017b; Messori et al., 2017; Hochman et al., 2019). The metric $\alpha$, which we term *co-recurrence ratio*, is a measure of the dynamical coupling between two variables, independently of their values (e.g. wet or dry), or in other terms their dependence structure.

The calculation of the dynamical systems metrics stems from the combination of Poincaré recurrences with extreme value theory (Lucarini et al., 2012; Freitas et al., 2010; Faranda et al., 2020). By *recurrences* we refer to the system being analysed returning arbitrarily close to a previously visited state in the phase-space. Given an atmospheric variable $x$, we consider a state of interest $\zeta_x$. In our case, this would be an instantaneous configuration of that variable, such as a latitude-longitude temperature map on a given day over the MED. We then consider recurrences to be those states that are close to $\zeta_x$, namely other timesteps at which the selected variable takes a very similar configuration. In order to quantify how close two configurations are to one another, we use the Euclidean distance (*dist*) between latitude-longitude maps. To compute the recurrences we first define an observable via logarithmic returns as follows:

$$g(x(t),\zeta_x) = -\log[\text{dist}(x(t),\zeta_x)] \tag{1}$$

Where $x(t)$ represents the time-series of $x$. We then define a threshold $s(q,\zeta_x)$ as a function of high $q$-th quantile of the time-series $g(x(t),\zeta_x)$. Next, $\forall\, g(x(t),\zeta_x) > s(q,\zeta_x)$ we define an exceedance $u(\zeta_x) = g(x(t),\zeta_x) - s(q,\zeta_x)$. The cumulative

probability distribution $F(u, \zeta_x)$ then converges to the exponential member of the Generalized Pareto Distribution (Freitas et al., 2010; Lucarini et al., 2012):

$$F(u, \zeta_x) \simeq \exp \left[ -\vartheta(\zeta_x) \frac{u(\zeta_x)}{\sigma(\zeta_x)} \right] \qquad (2)$$

Where $\vartheta$ is the extremal index (Moloney et al., 2019), and we estimate it here following Süveges (2007). The dynamical systems persistence is computed as: $\theta^{-1}(\zeta_x) = \Delta t / \vartheta(\zeta_x)$. In our case, $\Delta t = 1$ day and $\theta_x^{-1}$ has the units of the timestep of the data being analysed (i.e. *days*). For conciseness, we hereafter adopt the notation $\theta_x^{-1}$ to refer to the persistence of state $\zeta_x$.

To extend the analysis to two variables, $x$ and $y$, we can compute joint logarithmic returns around a state of interest $\zeta = \{\zeta_x, \zeta_y\}$ as follows:

$$g(x(t), y(t)) = -\log \left[ \text{dist} \left( \frac{x(t)}{\|x\|}, \frac{\zeta_x}{\|x\|} \right)^2 + \text{dist} \left( \frac{y(t)}{\|y\|}, \frac{\zeta_y}{\|y\|} \right)^2 \right]^{\frac{1}{2}} \qquad (3)$$

Where $\|.\|$ represents the average root mean square norm of a vector's coordinates. Once joint logarithmic returns are defined, we compute the co-persistence $\theta_{x,y}^{-1}$ based on the recurrences around $\zeta$. This effectively amounts to a weighted average of $\theta_x^{-1}$ and $\theta_y^{-1}$ (Faranda et al., 2020; Abadi et al., 2018). In our analysis, the joint state $\zeta = \{\zeta_x, \zeta_y\}$ would correspond to two instantaneous latitude-longitude maps: one for precipitation and one for temperature.

We further define the co-recurrence ratio (Faranda et al., 2020) $\alpha$ between $x$ and $y$ as:

$$\alpha(\zeta) = \frac{\nu[g(x(t)) > s_x(q) | g(y(t)) > s_y(q)]}{\nu[g(x(t)) > s_x(q)]} \qquad (4)$$

Where $s_x(q)$ and $s_y(q)$ are high $q$-th quantiles (or thresholds) of the univariate logarithmic returns $g(x(t))$ and $g(y(t))$, and $\nu[-]$ represents the number of events that satisfy condition $[-]$. Given a state $\zeta = \{\zeta_x, \zeta_y\}$, the co-recurrence ratio $0 \leq \alpha \leq 1$ measures the number of events where $x$ resembles $\zeta_x$ given that $y$ resembles $\zeta_y$, versus the number of cases when only $x$ resembles the relevant reference state. When $\alpha = 0$, there are no co-recurrences of $\zeta = \{\zeta_x, \zeta_y\}$ when we observe a recurrence of $\zeta_x$. When $\alpha = 1$, recurrences of $\zeta_x$ are always also co-recurrences of $\zeta = \{\zeta_x, \zeta_y\}$. Hence, $\alpha$ may be interpreted as a measure of the dynamical coupling between $x$ and $y$. However, $\alpha$ does not indicate causality: indeed, the order of $x$ and $y$ may be exchanged without affecting the value of $\alpha$.

In order to compute the dynamical metrics we use a quantile $q = 0.98$ to determine $s$. In previous studies (e.g., Faranda et al., 2011; Lucarini et al., 2012; Faranda et al., 2017b, 2019), this value has provided good estimates of the dynamical indicators, as it is high enough to select only genuine recurrences of $\zeta$, while also ensuring a sufficiently large sample of recurrences for analysis. Tests further showed little sensitivity of the results to $q$ in the range $0.95 < q < 0.99$ (Faranda et al., 2017b).

Finally, the dynamical systems approach rests on a number of theoretical assumptions, not all of which are strictly fulfilled by climate data. Specifically, the framework assumes the existence of an underlying chaotic attractor for the dynamics, and was derived for ergodic systems (Freitas et al., 2010). However, recent applications have shown that weak nonstationarities do not preclude the validity of the results (e.g., Faranda et al., 2019), provided that they do not lead to bifurcations of the system. Unlike common statistical techniques (e.g. copulas), which rely on extrapolation of extreme values from statistical distributions, the metrics we use here are grounded in the underlying dynamics of the system being analysed.

In our analysis, we consider each daily timestep in our datasets in turn as the state of interest $\zeta$. The final result of our analysis is therefore a value for each metric and timestep for the MED domain. This allows us to relate specific values of the metrics to the corresponding geographical anomaly patterns. We term *compound dynamical extremes* (CDEs) the days characterised by $\alpha > 90^{th}$ quantile of the full-year distribution over the 1979-2018 period. We selected the $90^{th}$ quantile as a good balance between an extreme value threshold and obtaining a sufficiently large sample of events. As sensitivity test we repeated the analysis in Section 4.2 for a $95^{th}$ quantile threshold, obtaining similar results (not shown). The two dynamical metrics successfully reflect large-scale features of atmospheric motions, and have recently been applied to a range of different climate variables over different geographical domains (Faranda et al., 2017a, b, 2019, 2020; Messori et al., 2017; Rodrigues et al., 2018; Hochman et al., 2019, 2020; De Luca et al., 2020; Scher and Messori, 2018).

## 2.2 Data

We use the European Centre for Medium-Range Weather Forecasts (ECMWF) ERA5 reanalysis over 1979-2018, with a spatial horizontal resolution of 0.25° and a 6-hourly temporal resolution (C3S, 2017). Our MED domain follows the "Full Mediterranean" region described in Giorgi and Lionello (2008). For ERA5, this corresponds to 27.75–48.00 °N, 9.75 °W–39.00 °E. To improve the robustness of our results, we have repeated the bulk of the analysis on ERA-Interim (Dee et al., 2011) and ERA5 10-member ensemble (C3S, 2017) (see Supplementary Material). We use the instantaneous 6-hourly data to compute daily maximum and minimum 2m temperature (K) and forecasted 1-hourly data for daily total precipitation (mm), from now on termed Tmax, Tmin and P respectively. Warm-dry days are days displaying positive Tmax and negative P anomalies relative to JJA means. Similarly, cold-wet days are DJF days displaying negative Tmin and positive P anomalies relative to DJF means. These are collectively referred to as 'compound events' and the corresponding anomaly means are computed individually at grid-point-level. Therefore, if for example a grid-point in a given day is warm it does not necessarily imply that it is also dry. We also analyse daily-mean sea-level pressure (SLP, hPa) anomalies relative to JJA (DJF) means, computed from instantaneous 6-hourly steps.

## 2.3 Statistical tests

The statistical significance of the Sen's slopes (Sen, 1968) of the $\alpha$ and $\theta^{-1}$ time-series is verified using the Mann-Kendall test (Mann, 1945) from the R package *'modifiedmk_v1.4.0'*. The Sen's slopes provide information about the steepness of the trends. If the Sen's slope is positive (negative) the corresponding trend is increasing (decreasing).

The statistical significance of SLP, Tmax, Tmin and P composite anomalies occurring during CDEs is computed using a one-tailed Mann-Whitney test at the 5% confidence level (Mann and Whitney, 1947). The null hypothesis is that a randomly selected median anomaly value during a CDE is equally likely to be less than or greater than a randomly selected median value from the days that are not CDEs. The alternative hypothesis is that during JJA (DJF), the SLP and Tmax (Tmin) median anomalies observed during CDEs are higher (lower) than those observed during other days. For P in JJA (DJF), the alternative hypothesis is that anomalies observed during CDEs are lower (higher) than those during other days. To avoid incurring in Type I errors (or false positives), we apply the Bonferroni correction to all p-values when considering single-gridpoint data (Bonferroni, 1936). The one-tailed Mann-Whitney test is also applied to the cumulative distribution functions (CDFs) of the anomaly means occurring during CDEs versus all other days.

Lastly, we checked the statistical significance of the percentage (%) agreement between JJA (DJF) CDEs and compound events. Here, the null hypothesis is that the JJA (DJF) observed % agreement is due to chance and to compute the significance the following steps have been followed: i) create n=1,000 datasets of random dates, with the same number of elements in each dataset as we have for the CDEs; ii) compute the % of agreement between CDEs and compound events' days for each dataset and grid-point; iii) pool together all the random % values and compute their $1^{st}$ and $99^{th}$ quantiles for each grid-point; iv) check whether the observed % values fall outside the random quantile values, and if this is the case consider the % values statistically significant at the 1% level (p-value <0.01).

## 3 Temperature-precipitation coupling

During JJA, the co-recurrence ratio ($\alpha$) between Tmax and P shows a significant upward trend (p-value <0.01) over 1979-2018 (Figures 1a and S1a). This points to an increasingly strong coupling between Tmax and P over time. Similar trends are also obtained when considering Tmin and P (not shown). During DJF, we also observe positive, albeit non-significant, $\alpha$ trends for all three reanalysis products (Figure S2). There is a clear correlation between $\alpha$ and summer mean Tmax, as highlighted in Figures 1b and S1b. Indeed, ranking $\alpha$ values by JJA-averages of Tmax results in positive and statistically significant trends (p-values <0.01), comparable in magnitude to those seen in Figures 1a and S1a. Moreover, both a regression analysis and the two-sided Spearman's rank correlation test (Corder and Foreman, 2014) between JJA $\alpha$ values and JJA average Tmax over the MED show a clear association between them (Figure S3). Trends in the $\alpha$ time-series of both CDE and non-CDE days are positive and statistically significant (Figure S4), pointing to a general shift in the $\alpha$ distribution towards higher values.

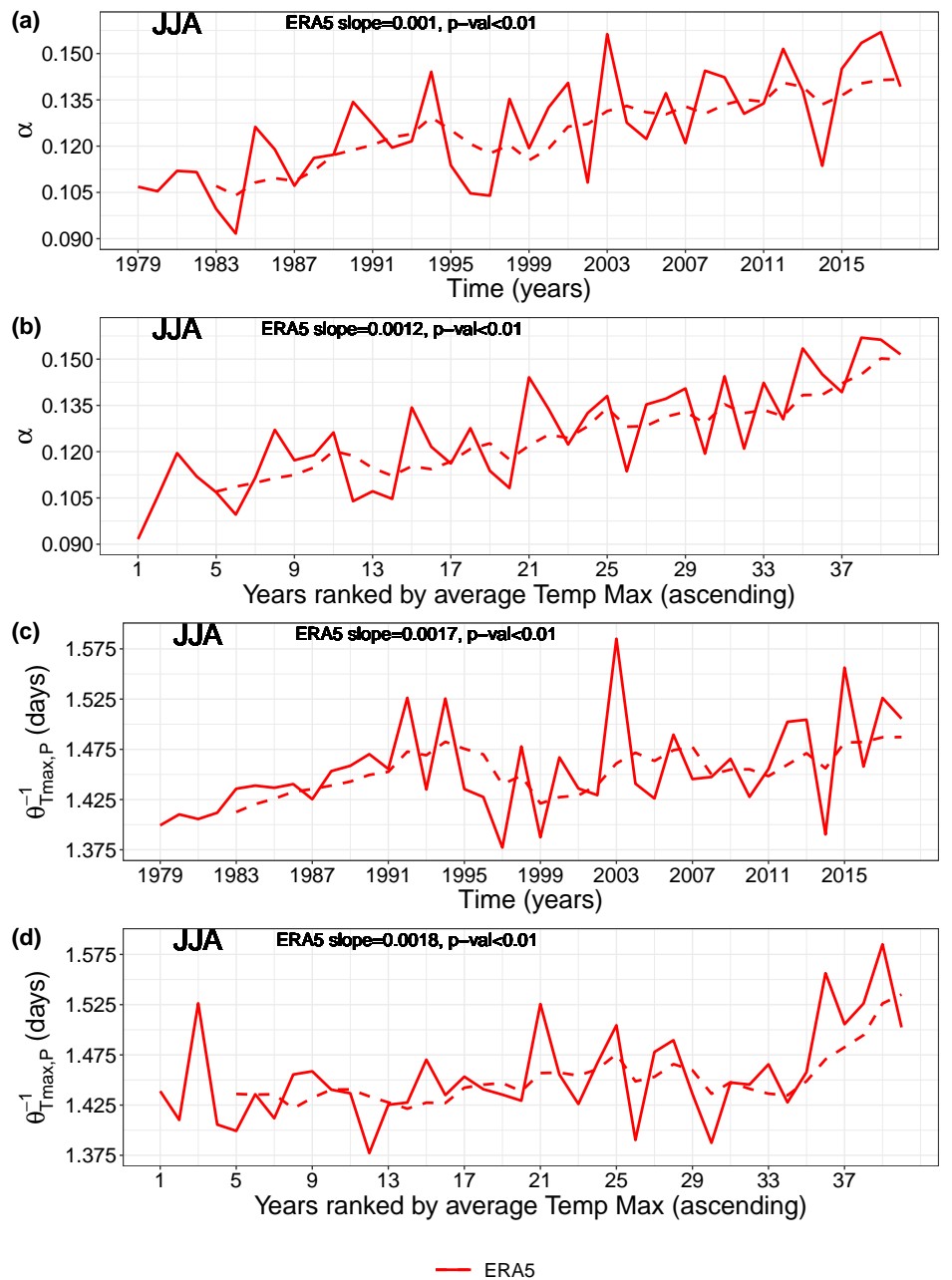

**Figure 1.** Co-recurrence ratio ($\alpha$) and local co-persistence $\theta_{Tmax,P}^{-1}$ JJA means for ERA5 during the 1979-2018 period over the Mediterranean (MED). (a) $\alpha$ JJA yearly means; (b) $\alpha$ ranked according to ascending JJA average Tmax; (c) $\theta_{Tmax,P}^{-1}$ JJA yearly means; and (d) $\theta_{Tmax,P}^{-1}$ ranked according to ascending JJA average Tmax. The dashed lines are 5-year centered moving averages. The Sen's slopes and p-values are also shown.

We next compute the local co-persistence ($\theta_{Tmax,P}^{-1}$) trends during JJA (Figures 1c and S1c) in analogy to Figures 1a and S1a. The significant upward trends (p-value <0.01 for ERA5 and ERA-Interim, and p-value <0.05 for ERA5 ensemble) in

$\theta_{Tmax,P}^{-1}$ imply a trend towards longer-lasting joint spatial patterns of Tmax and P over the MED within the observational period. By computing the co-persistence trends with only warm-dry days, similar results are obtained (not shown), pointing towards increasingly long warm-dry events over the region. As for $\alpha$, changes in co-persistence map directly onto changes in average Tmax in JJA (Figures 1d and S1d). Interestingly, there is a clear peak in $\theta_{Tmax,P}^{-1}$ during summer 2003 for all reanalysis products, coinciding with the extreme 2003 European heatwave (Black et al., 2004; Fischer et al., 2007; Stott et al.,

2004). Moreover, similar trends as for Figure 1 are obtained when computing $\alpha$ and $\theta_{Tmax,P}^{-1}$ for land-only grid-points (Figure S5a-d). The same, albeit with lower values, applies for $\alpha$ trends over sea-only (Figure S5e-f), while $\theta_{Tmax,P}^{-1}$ in this case does not show statistical significance (Figure S5g-h). The latter may be related to the damping role of the sea on air temperatures, although a more systematic analysis would be required to ascertain this. The trends in $\theta_{Tmax,P}^{-1}$ reflect trends in the (univariate) local persistence of Tmax ($\theta_{Tmax}^{-1}$) and P ($\theta_P^{-1}$) (Figures 2 and S6). They also at least in part explain the trends in $\alpha$, since

one may intuitively expect a higher co-persistence to lead to a higher co-recurrence ratio. We indeed find that $\theta_{Tmax,P}^{-1}$ and $\alpha$ are positively and significantly correlated in JJA (not shown). Trends in $\theta_{Tmax}^{-1}$ (Figures 2a and S6a) are stronger than those in $\theta_P^{-1}$ (Figures 2b and S6b). This strengthens our interpretation of Tmax as playing a predominant role in setting the observed positive trends in the dynamical metrics.

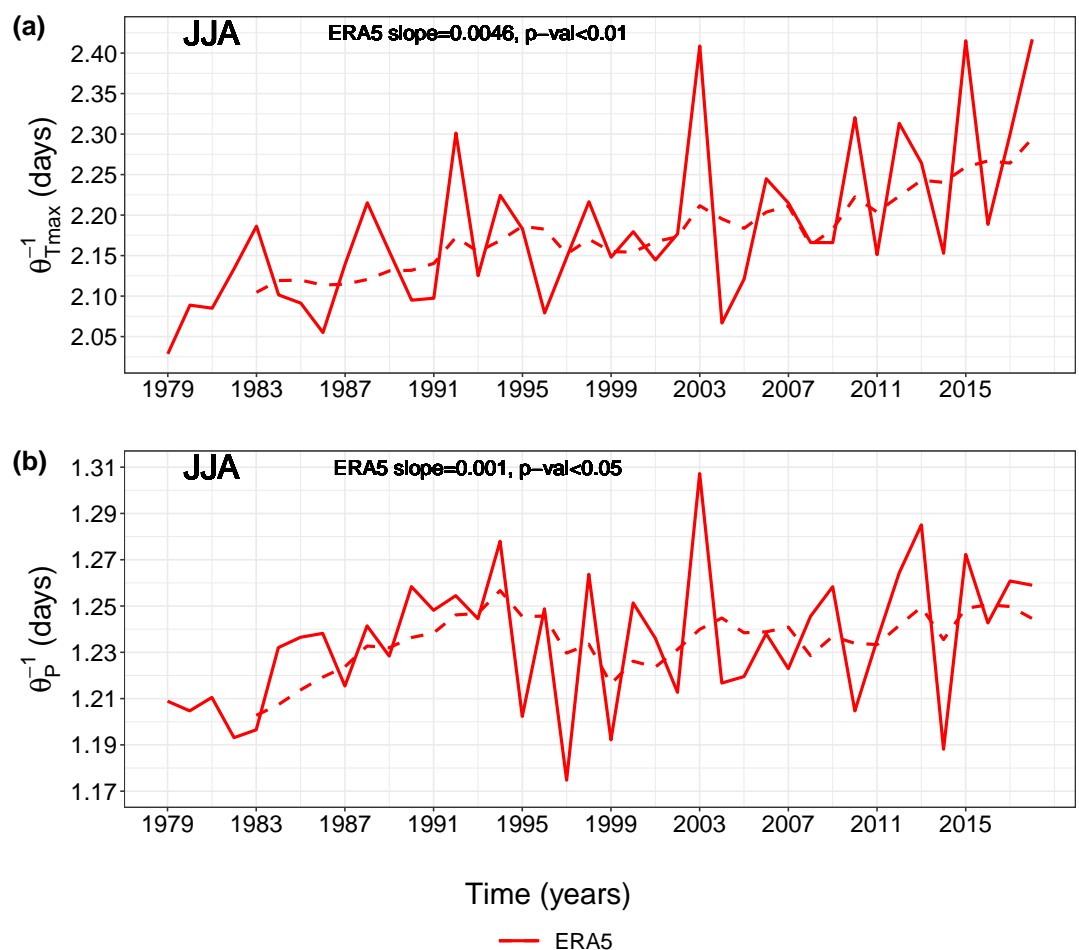

**Figure 2.** As Figure 1c but for univariate local persistence of (a) Tmax ($\theta_{Tmax}^{-1}$) and (b) P ($\theta_P^{-1}$).

# 4 Compound dynamical extremes (CDEs) linked to compound warm-dry and cold-wet events

## 4.1 Seasonality of CDEs

We next investigate the temporal distribution of CDEs. For $\alpha$ computed on Tmax and P, all three reanalysis products display most of the CDEs clustering in July and August, with a secondary peak in DJF (Figures 3a and S7a). For $\alpha$ computed on Tmin and P, most CDEs occur during DJF, July and August (Figures 3b and S7b). This holds for all three reanalysis products. We hypothesise that the large number of CDEs during July and August (Figures 3b and S7b) can be linked to extreme summertime precipitation events, that cool the air and increase wetness (e.g., Stadtherr et al., 2016; Christensen and Christensen, 2003). We further note that, notwithstanding the previously mentioned correlation between co-persistence and alpha, the seasonality of $\theta^{-1}_{Tmax,P}$ extremes – defined analogously to the CDEs – does not reflect that of the CDEs (not shown). For both variable combinations, the two shoulder seasons (i.e. spring and autumn) display very few CDEs. In Faranda et al. (2017a), the authors hypothesised that during autumn and spring the atmospheric flow sits on a saddle-like point of the dynamics, while winter and summer represent more stable basins of attraction. Assuming that distinct attractors indeed exist for winter and summer, we thus interpret these low CDE counts as the result of the atmospheric flow exploring both summer and winter configurations, resulting in rarer co-recurrences.

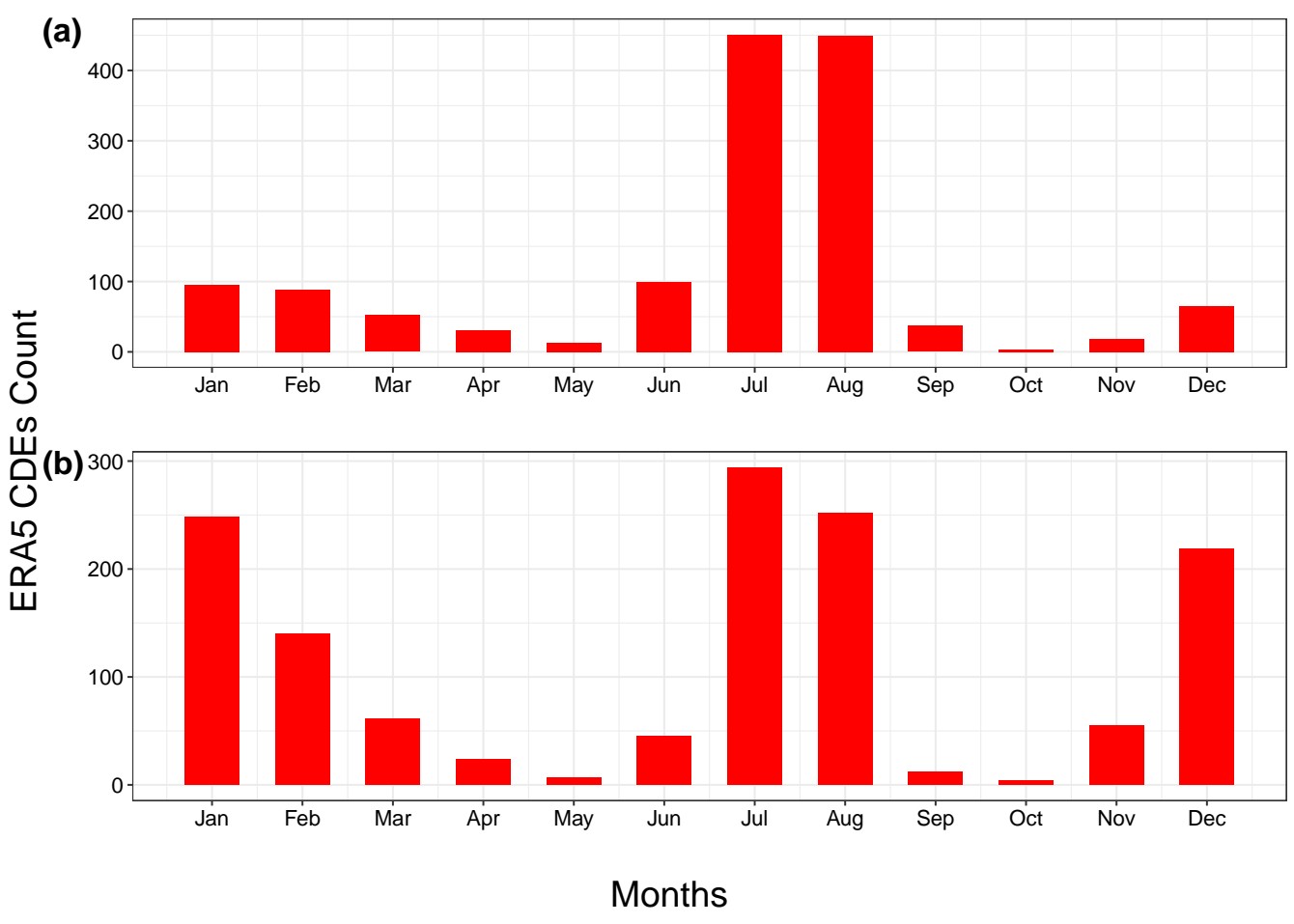

**Figure 3.** Monthly counts of compound dynamical extremes (CDEs) for ERA5 during 1979-2018 over the MED. (a) $\alpha$ computed from Tmax and P; and (b) $\alpha$ computed from Tmin and P. CDEs are defined as $\alpha$ daily observations $> 90^{th}$ quantile of the $\alpha$ distribution for the full dataset.

## 4.2 Pressure, temperature and precipitation anomalies during CDEs

During JJA, CDEs correspond to statistically significant positive SLP anomalies over the western MED (north-western Africa) and the Anatolia - Black Sea region. These are separated by negative SLP anomalies spanning the Aegean sea, the Levant and Northern Egypt (Figures 4a and S8a-b). These SLP anomalies are in turn associated with significant warm Tmax anomalies over most of the MED, with a particularly warm Balkan Peninsula, and a negative anomaly over central northern Africa, to the east of the positive SLP anomaly (Figures 4c and S8c-d). Lastly, we observe weak dry P anomalies over the Black Sea (Figures 4e and S8e-f) and stronger wet P anomalies over the Alps. The latter correspond to statistically significant convective available potential energy (CAPE, $JKg^{-1}$) positive anomalies (Figure S9), and may therefore be linked to localised convective P events. We conclude that JJA CDEs are closely linked to widespread warm Tmax anomalies, but have a weaker footprint on P anomalies, except over the Alps.

In DJF we observe an east-west dipole in SLP over the MED, that favours cold-air advection from northern Europe to the Balkans, parts of the Italian Peninsula and the Southern and Eastern MED (Figure 4b and S10a-b). Indeed, negative and significant Tmin anomalies are observed over most of the MED region (Figure 4d and S10 c-d). The Eastern MED also displays significant positive P anomalies (Figure 4f and S10e-f). The statistically significant (p-value <0.05) negative SLP anomalies over the Eastern MED are reminiscent of the footprint of Cyprus Lows, which are the main rain-bearing systems over the region (Alpert et al., 2004; Saaroni et al., 2010) (Figures 4b and S10a-b). Cyprus Lows are also associated with the majority of wintertime cold spells over the Eastern MED (Hochman et al., 2020), and we indeed find that some of the P anomalies over the Eastern MED are snowfall events, particularly over the Balkans, Turkey and Lebanon (Figure S11). We thus conclude that CDEs are associated with wintertime cold-wet compound events over the Eastern MED.

As a proxy for the variability within our composites in Figure 4, we compute the standard deviations (SDs) of the anomalies (not shown). We observe that SLP SDs are larger over the northern and central MED, while temperature SDs are larger over land compared to the sea – the latter a natural consequence of the sea's large thermal inertia. Finally, precipitation SDs are larger where the higher anomaly mean values are reported (i.e. the Alps in JJA and south-Eastern MED in DJF), which may be linked to the prevailingly dry summertime conditions in the MED which yield low SDs where little or no rain falls. Similar results are obtained when computing Figure 4 using only extreme anomalies ($anom > 90^{th}$ and $anom < 10^{th}$ quantiles) matching CDEs, although the JJA positive SLP anomalies are less geographically extensive (Figure S12).

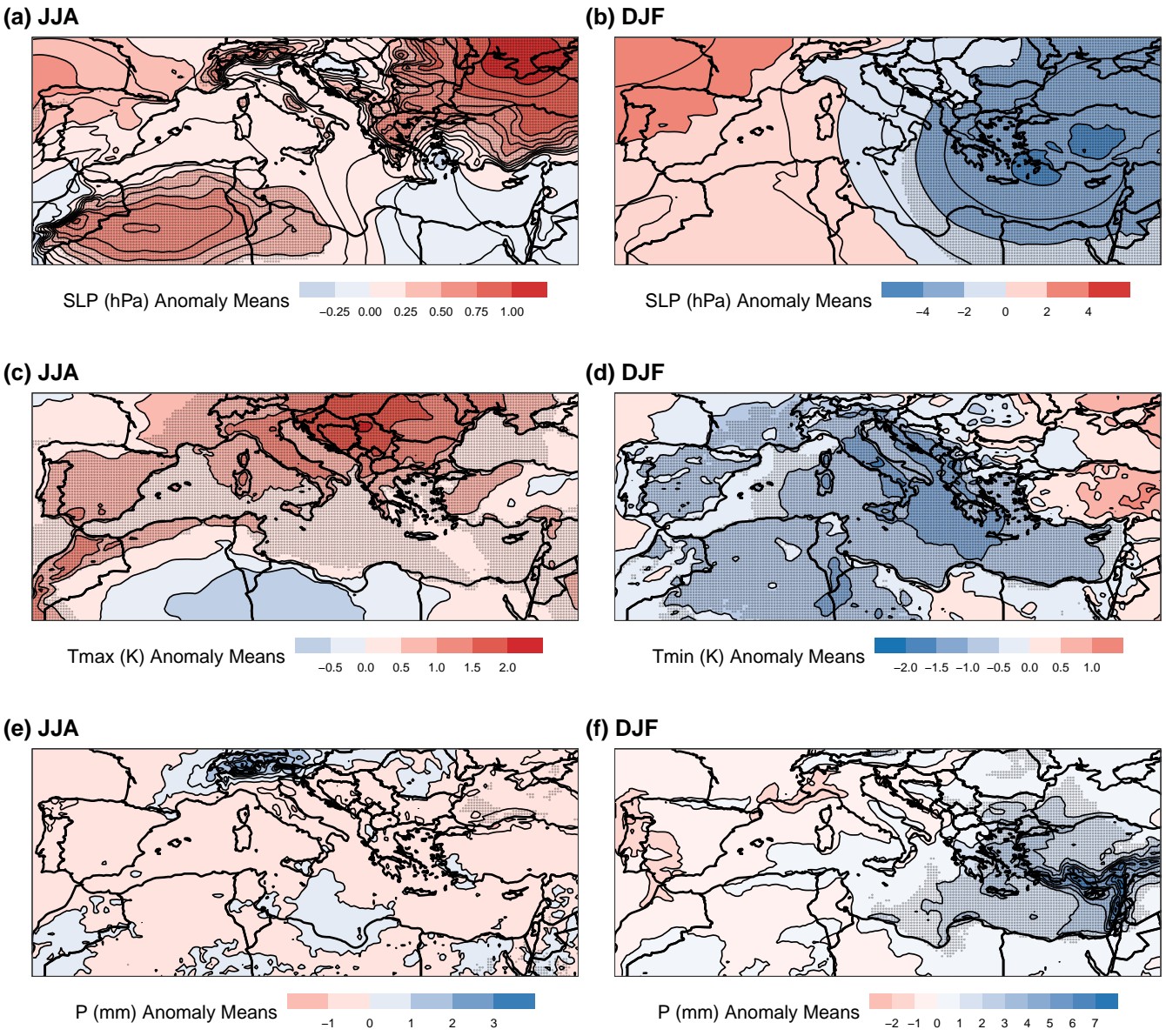

**Figure 4.** JJA and DJF anomaly means of (a-b) SLP, (c-d) Tmax and Tmin, and (e-f) P during CDE days. The data are from the ERA5 reanalysis during 1979-2018. $\alpha$ for JJA is computed from Tmax and P, whereas for DJF from Tmin and P. Stippling shows statistically significant anomalies (p-value <0.05, Mann-Whitney one-tailed test). The Bonferroni correction is applied to all p-values.

### 4.3 Distributions of temperature and precipitation anomaly means

We next test empirically whether the CDEs highlighted above have a systematic link to compound JJA warm-dry and DJF cold-wet events. During JJA, Tmax and P daily anomaly means, computed for each grid-point during CDEs, are predominantly warm (85%) and dry (79%) respectively (Figure 5a-b). Similar results are also obtained for ERA-Interim and ERA5 ensemble (Figure S13). P anomalies tend to cluster around zero, owing to the overall dry summertime climate of the region, although as noted above they do show a preference for negative (dry) values (Figures 5b and S13b,d). A Mann-Whitney one-tailed test

between the anomaly means during CDEs versus all other days in JJA results in statistically significant differences (p-value $\ll 0.01$) for all reanalysis products for both Tmax and P. This implies that CDEs are significantly warmer and drier than other JJA days.

  In DJF, most of the Tmin and P anomaly means are cold (78%) and wet (58%) respectively for ERA5 (Figure 5c-d) and the other reanalysis products (Figure S14). Again, a Mann-Whitney one-tailed test between anomaly means during CDEs and

all other DJF days highlights statistically significant (p-value $\ll 0.01$) differences for all reanalysis products', except ERA-Interim's P (p-value $<0.05$). This implies that CDEs are significantly colder and wetter than all other DJF days. The CDEs therefore present a somewhat mirror image of the preferred anomalies seen in the geographical anomaly composites for both JJA and DJF.

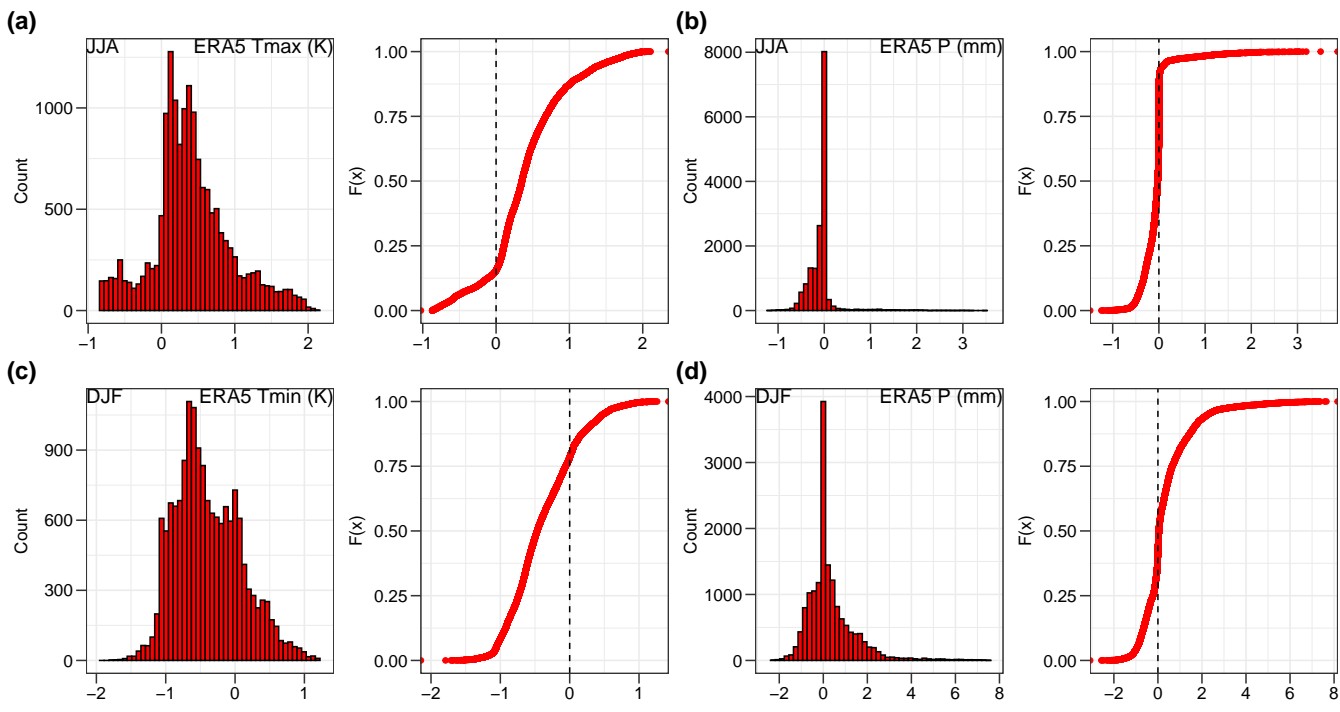

**Figure 5.** Histograms and cumulative distribution functions (CDFs) of anomaly means of (a) Tmax, (b) P during JJA CDEs, and (c) Tmin, (d) P during DJF CDEs. The data are the same as in Figure 4c-f. The distributions are statistically different from those of all other JJA and DJF days, respectively (p-value ≪0.01, Mann-Whitney one-tailed test).

## 4.4  Spatial patterns of compound warm-dry and cold-wet events

We next complement the statistical information provided by the histograms and CDFs with spatial distributions of percentage (%) match between CDEs and compound events. Simply, for each grid-point in Figure 6 we identify the days reporting compound events *and* CDEs, then divide the total number of these days by the total number of CDEs and multiply the resulting number by 100 to obtain the % agreement value. Across the MED, a high fraction of CDEs coincide with compound warm-dry events during JJA. Values locally exceed 70%, meaning that >70% of all JJA CDEs occur during compound warm-dry events

(Figures 6a and S15). The highest percentages occur in southern Spain, the Balearic Islands, Italy and the Balkans. During DJF, the % match between CDEs and compound cold-wet events is lower than that seen for warm-dry JJA events (<50%) (Figures 6b and S16). The highest % occurs over the Eastern MED sea, between the coastlines of Libya, Egypt, Greece and Turkey. In both JJA and DJF the vast majority of observations (%) are statistically significant at the 1% level (p-value <0.01, Figures 6, S15-S16).

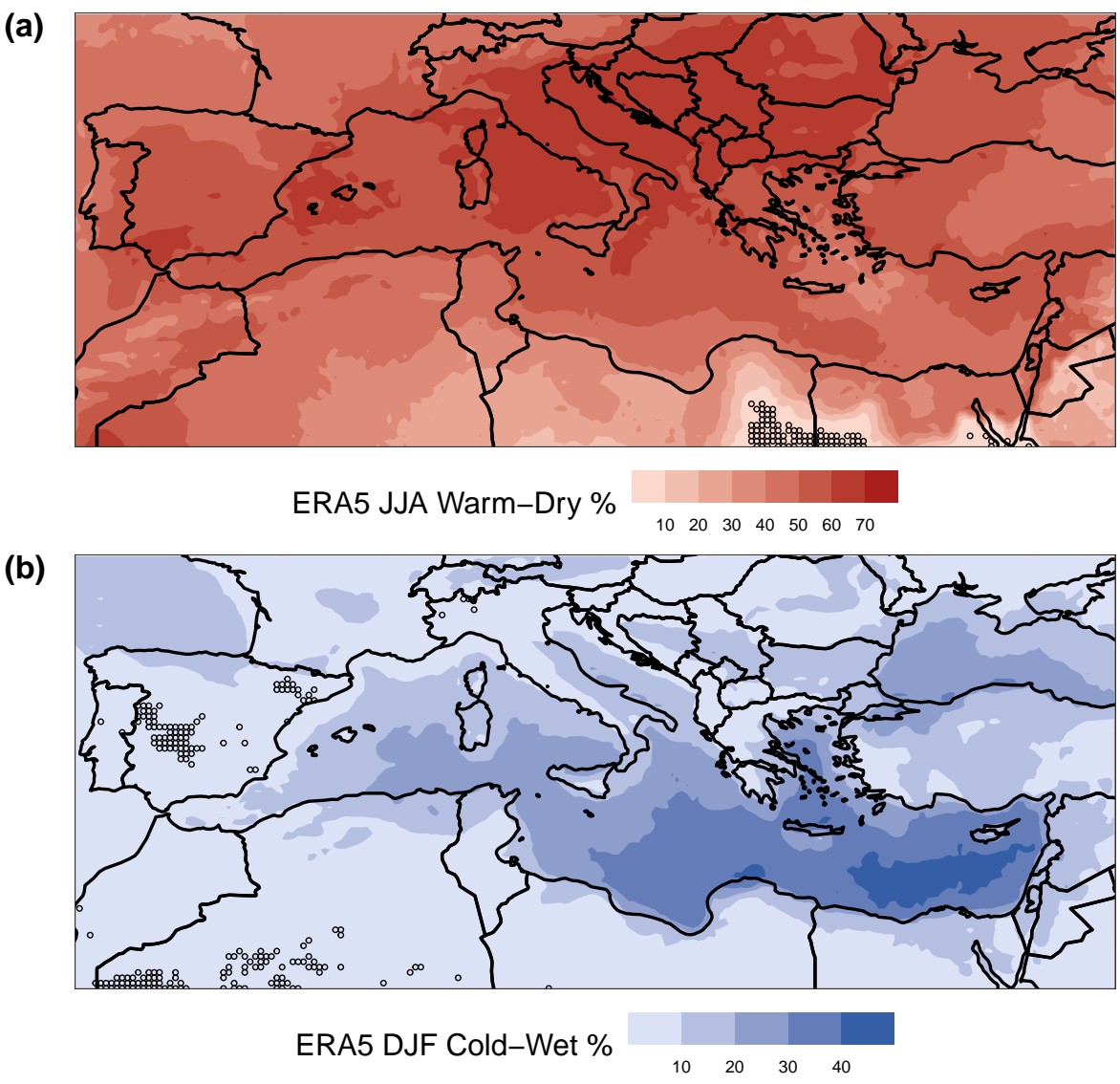

**Figure 6.** Percentage (%) of CDEs occurring during compound (a) JJA warm-dry and (b) DJF cold-wet events. The data are from the ERA5 reanalysis during 1979-2018. Stippling represent values *not* statistically significant at the 1% level (p-value ≥ 0.01).

## 5 Discussion and conclusions

In this paper, we analysed compound warm-dry (cold-wet) events during JJA (DJF) over the Mediterranean (MED) through the lens of dynamical systems theory. We specifically computed a measure of coupling ($\alpha$) between daily maximum temperature (Tmax) and total precipitation (P) during JJA and daily minimum temperature (Tmin) and P during DJF. We then identified days when the two variables are strongly coupled ($\alpha > 90^{th}$ percentile of its full distribution) and termed them compound dynamical extremes (CDEs). We further computed a dynamical systems measure of the persistence of large-scale configurations in the above variables ($\theta^{-1}$), considering them both individually and in pairs. We made use of the ERA5 dataset but replicated the analyses also with ERA-Interim and ERA5 10-member ensemble (see Supplementary Material). We generally found a good agreement between the different reanalysis products.

During JJA, both $\alpha$ and $\theta^{-1}_{Tmax,P}$ display significant upward trends. An upward persistence trend is also found if we focus specifically on warm-dry days. We propose these trends are driven by surface warming over the MED. A possible physical process driving increasing coupling with increasing temperature is soil drying. Although we didn't investigate this in detail here, we found that also a decrease in average P is linked with an upward and significant trend in $\alpha$ (Figure S17) and that the correlation between Figures 1b and S17 $\alpha$ values is positive and significant ($\rho$=0.56, p-value <0.01). Specifically, the increasingly warm summer temperatures and lack of P may lead to significantly lower soil-moisture content, triggering a feedback mechanism that favours persistent warm-dry conditions. However, at this stage, is difficult to discern between the prevailing role between Tmax and P in driving the $\alpha$ trends, since they may have a compound or univariate effect. We will therefore keep this investigation for a further work. Consistently with the $\alpha$ trends, we found that CDEs computed from Tmax and P cluster during July and August, whereas CDEs computed from Tmin and P cluster during July, August and DJF. During CDE days, synoptic patterns in JJA show significant positive SLP and warm Tmax anomalies over large parts of the MED, and dry but mainly not-significant anomalies for P. The latter is somewhat unsurprising, as the low climatological summertime precipitation over the region effectively prevents the occurrence of large negative precipitation anomalies. Moreover, Tmax anomalies result stronger over land than over the sea, because the latter's thermal inertia likely plays a damping role during the occurrence of heatwaves. Lastly, the JJA SLP patterns do not point to any clear and documented synoptic structure. It may therefore be possible that CDEs capture several different sets of weather circulation regimes. In DJF, CDEs are associated with significant negative SLP anomalies and cold-wet anomalies centred over the Eastern MED. The distributions of anomalies occurring during CDEs are significantly different (p-values of <0.01 or <0.05) from the ones recorded during all other days. Lastly, we found that CDEs correspond to a heightened frequency of positive Tmax and negative P anomalies during JJA, and to a heightened frequency of negative Tmin and positive P anomalies during DJF over large parts of the MED. The percentages of CDEs matching cold-wet days during DJF are, however, lower than those found during summer for warm-dry days.

The findings that summertime Tmax and P have become more strongly coupled over the last 40 years, and that the persistence of warm-dry days has increased, are in agreement with Zscheischler and Seneviratne (2017) and Manning et al. (2019). The former study showed that land-atmosphere feedbacks in a warmer world may lead to an increase in warm-dry summers larger

than what may be expected by analysing the projected temperature and precipitation changes as single variables. However, the work of Zscheischler and Seneviratne (2017) differs from ours since they made use of detrended temperature and precipitation datasets. Whereas Manning et al. (2019) found that rising temperatures drive an increased probability of dry and hot events in Europe, with dry periods becoming hotter and hence pointing to a significant thermodynamic response of compound events due to global warming. Assuming a continued increase in future temperatures, we may therefore expect ongoing positive JJA $\alpha$ and $\theta_{Tmax,P}^{-1}$ trends, leading to a higher frequency of compound JJA warm-dry events.

The analysis of DJF CDEs, matching cold-wet events, points to very different dynamics. Here, the largest anomalies in SLP, Tmin and P are found over the Eastern MED, and are reminiscent of the footprint of Cyprus Lows. These are wintertime synoptic systems that play a predominant role in driving concurrent cold spells and heavy precipitation events over the Levant (e.g., Hochman et al., 2019). Our findings show no significant increase in $\alpha$ values during DJF, in line with studies suggesting a decrease in Cyprus Lows frequency, persistence and associated precipitation over the Eastern MED (Hochman et al., 2020, 2018; Peleg et al., 2015).

Our findings highlight a close connection between CDEs, computed from dynamical systems coupling, and compound JJA warm-dry and DJF cold-wet events over the MED. The link between CDEs and compound events likely issue from the fact that, in both cases, the data reflect anomalous (or highly-coupled) conditions for the atmospheric variables being studied. It is of particular interest that $\alpha$ distinguishes between JJA warm-dry and DJF cold-wet compound events. However, results obtained from our dynamical systems approach may be sensitive to the size and location of the geographical domain(s) under study. For such reason, it is important to constrain the dynamical systems analysis only over a geographical area justified by for example physical process understanding or impact assessment. In the latter case, one may be interested to calculate *compound climate risks* by making use of CDEs as a measure of the multi-hazard component or link $\alpha$ with (long-enough) impact datasets, such as insurance losses, crop yield or renewable energy production.

Based on our results, we learn the following: i) the coupling between temperature and precipitation at large scales is driven by specific regions and processes (e.g. Cyprus-low) and therefore it does not always reflect the whole MED; ii) the coupling results are sensitive even to non-extreme events, and thus the co-recurrence ratio ($\alpha$) may be fruitfully used in forthcoming studies to elucidate potential future seasonal climatic changes over the MED; and iii) our results provide information on specific factors that are driving the changes in $\alpha$ (e.g. surface warming). In the future, we envisage making use of global CMIP6 data under different Shared Socioeconomic Pathways (SSPs) up to 2100 (O'Neill et al., 2016) and abrupt climate change simulations (e.g. 4xCO2) (Eyring et al., 2016). These investigations may also shed some light on possible tipping points over the MED (Lenton et al., 2008; Lenton, 2011).

*Data availability.* The ERA5, ERA5 10-member ensemble and ERA-Interim reanalysis datasets used in this work are freely available from the European Centre for Medium-Range Weather Forecasts (ECMWF) websites *1* and *2*.

*Author contributions.* PDL designed the study, performed the analyses and created the figures. GM, DF and DC contributed to the methods
and study design. PDL and GM wrote the first manuscript draft. All the authors contributed to the writing.

*Competing interests.* The authors declare that they have no conflicts of interest.

*Acknowledgements.* This is TiPES contribution #15. This project has received funding from the European Union's Horizon 2020 research and innovation programme under grant agreement No. 820970. PDL was also supported by an E-COST STSM (DAMOCLES, Action CA17109). GM was partly supported by the Swedish Research Council Vetenskapsrådet under grant agreement No. 2016-03724. PJW was
pupported by a VIDI grant from the Dutch Research Council (NWO, grant nr: 016.161.324). The data analysis has been performed on the VU HPC BAZIS-cluster. The authors would like to thanks the three referees and editor for their constructive comments, which significantly improved the manuscript.

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
