# Peer review of "Compound Warm-Dry and Cold-Wet Events Over the Mediterranean"

_Earth System Dynamics, 2020_

## Referee Comment (RC1) · Emanuele Bevacqua (Referee) · 24 Apr 2020

**Review of the paper: "Compound Hot-Dry and Cold-Wet Dynamical Extremes Over the Mediterranean" by De Luca et al.**

**Reviewed by Emanuele Bevacqua**

The authors investigate compound hot-dry and wet-cold events over the Mediterranean basin, employing a novel method based on dynamical systems theory. They use different reanalysis products and find a tendency towards an increasing coupling between temperature and precipitation over 1979-2018. The paper is well written and pleasant to read. I find the use/introduction of this approach interesting given that it offers a novel perspective for studying compound events. New approaches are always welcome as they can challenge or confirm previous findings. I recommend publishing the paper, but also to address some comments that follow. All of the comments are, of course, meant be constructive.

A main comment I have is about the definition of hot&dry and wet&cold conditions. I understand that they are defined based on positive/negative seasonal anomalies of precipitation and temperature. Although also non-extreme values of the contributing variables can lead to extreme impacts, the employed anomalies may be be particularly small. Given that the authors link the study directly to compound events and associated risks in the Mediterranean area, I think that some considerations are required. The authors could repeat some of the analyses based on higher anomalies (see comment below). Alternatively, I would recommend modifying the text in several parts, including abstract (e.g., line 7) and title, to avoid giving the impression of referring to, e.g., hot (and therefore extreme) events. In general, when possible some more physical interpretation would be welcome to guide the reader.

**Specific comments**

L40, The paper from Manning et al. would definitely fit here (see also comment later): Manning C, Widmann M, Bevacqua E, Van Loon AF, Maraun D, Vrac M. Increased probability of compound long-duration dry and hot events in Europe during summer (1950–2013). Environmental Research Letters. 2019 Aug 29;14(9):094006.

L42, I would strongly suggest adding 2-3 sentences in this paragraph. You could explain, via examples and references, why wet-cold and dry-hot events can lead to impacts (e.g., wildfire, vegetation issues etc), i.e. why they are important.

From line 61 onward. Overall, the explanation is very easy to understand. However, would it be possible to add 1-2 equations to guide some type of readers?

L68, Does the persistence depend on any used threshold to define the close range dx (intorno) around Zeta_x? Or is it propriety of the system in Zeta_x that you somehow obtain based on some limits? Please, clarify.

L72, Also for the co-recurrence ratio: is this obtained based on (empirical) counting of the states and therefore it depends on the values dx and dx used to the define the close range

around Zeta_x and Zeta_y? I agree that a full description of the theory should not be given but, in my opinion, a few sentences to guide the reader are needed here.

"Compound dynamical extremes", would it be better to use compound dynamical events? "Extremes" might be misleading.

L92, anomalies relative to JJA means

L96, Could you please explain why these slopes are preferred to usual linear regression. For example, would the trend in Fig. 1c-d be non-significant with a linear regression? Please, discuss this.

L110, Fig. 1a, Are you computing alpha for every JJA day and then computing the yearly average? Make this clearer, please (in the caption is not fully clear in my opinion).

L110, Guiding the reader to see what is happening in the (T,P) space would help here. For example, in summer, would you expect to find a similar trend using Tmin and P (or, in winter, Tmax and P)?

L112, Would in any way carrying out the analysis after detrended the time series of the temperature help to better understand the physical drivers of the trends?

L 120, is there a correlation between co-persistence and alpha?

L121, how are, in this regional case, hot and dry days subsampled?

Section 4.1, Do also the univariate persistences show a similar seasonality? If so, is it possible to interpret this in relation to the seasonality in alpha?

Section 4.2 and the following sections. The authors could consider whether moving these results before the trends would help or not. Being aware of what alpha depicts from a physical point of view may help to interpret the trends more easily.

L141 (paragraph)

1) In general, a discussion on why one would expect to capture anomalies would be important to help the reader.
2) I understand that the anomalies are computed relative to the seasonal mean, please specify it.
3) To strengthen the conclusions, you could highlight that small anomalies are expected over the sea due to water inertia, even during heatwaves.
4) Overall, the SLP field picked up by high alpha values in winter appears associated with a more defined atmospheric configuration compared to summer. Is it possible that you pick up different weather circulations within the subsample of extreme alpha in summer?
   - I see you have stippling in Fig. 4. How large is the average anomaly during days with large alpha compared to the standard deviation*? *computed based on the daily anomaly data.

- How do these maps change when using, e.g., the 95th percentile to define extreme alpha?
5) L146, The latter correspond…: Personally, I would rephrase given that although the significant anomaly is all associated with the large-scale component of the precipitation, also the convective part is relevant. The reasoning in the next sentence would still work, maybe saying these anomalies are *mainly* linked to SLP.

L158, The sentence is correct. But it suggests that these events can occur everywhere over the analysed domain, while, especially the P anomalies, suggest that this is the case mainly over the eastern domain and along the Italian areas exposed to cold-air advection from northern Europe.

L160, I understand that hot-dry and cold-wet events are defined based on positive/negative anomalies from the seasonal average. Would the main conclusions be similar if using larger anomalies to define, hot/cold and wet/dry conditions? For example, one could use +/-2 standard deviations from 0 to define larger anomalies.

L167, Could you relate to the numbers above, i.g., does this also imply that also the values 84% and 77% are significantly large?

L162, How are these numbers computed: Are all CDE days and grid points pulled together? Please, clarify.
L168, consistently with Fig. 4f, the distribution of the precipitation is peaked around zero in fig. 5d. I am wondering whether (maybe for future work) the authors would see any added value in focussing some of these statistics only over the eastern part of the domain, where the framework is able to better capture anomalous conditions. If so, this could be discussed.

Section 4.4, It is a bit difficult to read the values in fig. 6 given that the palette has continuos values. Aren't these values depending on the percentiles (here 90th) used to define the CDE events? Therefore, the reader should be helped to interpret these numbers. They should be compared to what expected under a certain null hypothesis. For example, one could easily compute the probability of getting concurrent CDE and hot&dry days assuming that the CDE events are randomly distributed during the year (if this is a reasonable assumption).

Discussion: Could you add 1-2 sentences about the expected sensitivity of the results from the size of the analysed domain? This is relevant for the reader...

L190, Do you think that re-computing the trends in the two metrics obtained based on maps of (1) land surface only and of (2) sea surface only could somehow allow for speculating more safely about this? Or, more in general, could this allow for disentangling a higher signal of the increasing coupling on land?

I understand that the trends you found in the regional coupling can have different meaning depending on the areas of the domain so I understand that they should be interpreted bearing this in mind. Could you discuss more explicitly on this from an impact perspective?

Around L205, I would suggest to interpret and discuss the results also in relation to results found by Manning et al. (paper cited above).

L215, see my main comment about the definition of hot days.

The authors could consider expanding the discussion, very briefly, to highlights the potential benefits of their approach for the part of the compound event community that is focussing on impact assessments.

**More technical comments**

L10, discussing first the winter results may allow avoiding some repetitions.

L17, add space

L19, I would probably start talking of atmospheric circulation changes which are more intuitive than "dynamical changes" for a non-expert reader.

L20, may expect based on?

L27, an increase in "daily" or "episodic" precipitation extremes?

L56, "[...]. The metric theta^{-1}..."

L79, each daily timestep

L80, with -> characterized by…; L81, could be written slightly better

Fig.1, caption, L4, 5 year centered moving [...]

L200, Please, check that it is ok. I assume you did not talk of Tmax *and negative P* anomalies on purpose.

L 211, it is probably more correct to write "concurrent cold spells and heavy…"

L212, "e.g" in the parenthesis

L212, "accordance" a better word? It might not look in accordance with the decrease found int the other studies.

Well done. Best regards.

---

## Referee Comment (RC2) · Olivia Romppainen-Martius (Referee) · 14 May 2020

In this paper concurrent recurrence of temperature and precipitation patterns over the Mediterranean are quantified and changes over time highlighted. The paper is well written and the results are highly interesting and relevant. There are some methodological points that need further explanations for non-experts (explained in more detail under major points) and the colorbars of the map figures need to be improved. After these clarifications the paper is ready for publication. Olivia Martius

Major points: 1) Consider adapting the title because the term "dynamical extremes" may not be widely known. 2) I have some basic methodological questions that I did not yet fully understand and that might be confusing for other readers as well. a. Cooccurrence of dry and hot (cold and wet) conditions refers to the entire Mediterranean area. The dry conditions and the hot conditions must not necessarily happen at the same location in the Mediterranean area? b. You state that you get an alpha and theta value for each time step of the data, but it is unclear how long the time period is that you use to determine these parameters, you only mention a "relatively long time period" in the beginning. Is it a moving window of a number of days? Are then successive alpha and theta values very similar? c. Is it correct that compound dynamical extremes are "extreme" in the coupling but not necessarily in p and T? Do these CDEs then point to weather situations that are dominated by the synoptic-scale flow rather than by local convective systems? 3) Please use colorbars with discrete colors for all map figures. For example, when I am interested in the SLP anomaly over Italy in Figure 4a it is very difficult to link the discrete colors on the map to the continuous colors in the colorbar. 4) The interpretation of Figure 4 is difficult without a reference to the climatology. It would be good to indicate how the anomalies relate to the climatology (e.g. % of mean) and if feasible to the variability (e.g., STD, inter quartile range). For example, precipitation rates in the Alps are higher in the climatology. So maybe in Figure 4e the anomalies over the Alps are small from a climatological and variability perspective and the dry anomalies in other areas are large. This would also clarify a follow up question namely can your method capture dry and wet extremes at the same time? The discussion of Figure 4d focuses on the wet anomaly over the Alps but the subsequent discussion of the histograms points to the prevalent dry signal. This is confusing for the reader.

Minor points: 1) L16: Could you provide an example for the dynamical changes. 2) L28: Clarify what you mean by similar changes. 3) L44: Define large-scale precipitation, here I assume it is the model variable. 4) L47: Suggest to shift a couple of sentences from the next section up here to briefly explain what dynamical systems are. A dynamical meteorologist might think of weather systems at this point of the paper rather than system dynamics. 5) L92: How do you define anomalies? Are these to the 30-year (seasonally varying?) mean? 6) L92: But you also use large-scale vs. convective precipitation? Could you add the information that the precipitation is a

forecast field and not assimilated. 7) L121: what do you mean exactly be "restricting the analysis..."? 8) L141: Please define how you calculate the anomalies, wrt to a seasonal mean? 9) L141: Linking the precipitation anomalies with the SLP anomalies should be formulated in the form of a hypothesis. 10) L147: Please add supporting references. For heavy precipitation along the western Alpine south-side the low pressure system is typically located over the Gulf of Genoa. Also these types of low pressure systems are associated with cold fronts and colder surface temperature. However, the temperature pattern shows high temperatures in this area. An alternative interpretation is that the low pressure system over the Balkans might correspond to a heat low. It is unclear for me which aspects of the SLP distribution are related to the precipitation field and which ones to the temperature field. Can you separate this? This might also link back to major point 4. 11) L151: How does the cold air from northern Europe cross the Alps into Italy? This statement does not have any supporting analyses in the paper. Please either refer to the literature, show the trajectories or remove the statement. 12) L162: how do you define dry? 13) Please add units to Figure 5. 14) L175: It is unclear to me how you compute these maps since I understood the measures to be linked to one pattern over the entire Mediterranean. Please expand your explanations. 15) L195: weaker anomalies -> can you be more specific?

---

## Referee Comment (RC3) · Vera Melinda Galfi (Referee) · 1 Jun 2020

Dear de Luca et al.,

please find below the review of your manuscript.

Best regards, Melinda Galfi

GENERAL COMMENTS

The authors analyse hot-dry as well as cold-wet dynamical extremes over the Mediterranean region in ERA reanalysis data sets over the period 1979-2018. They use a novel method based on dynamical systems metrics and extreme value theory to select and analyse so-called "compound dynamical extremes." The study is mainly based on

two indicators, termed the co-recurrence ratio $\alpha$ and the co-persistence $\theta\hat{}(-1)$. They estimate these indicators for joint occurrences of daily maximum temperature and total precipitation, as well as of daily minimum temperature and total precipitation. They define the events with $\alpha$>90th quantile of the whole $\alpha$ distribution as compound dynamical extremes. The authors find a positive trend in the co-recurrence and co-persistence of hot-dry events during summer (JJA), whereas no trend can be found in case of the co-recurrence of cold-wet events in winter (DJF). Thus, they conclude that long-term warming strengthens the coupling between temperature and precipitation, leading to more intense hot-dry compound events. They also analyse spatial fields of sea level pressure, temperature and precipitation during compound dynamical extremes, as well as spatial maps of the co-recurrence ratio $\alpha$.

The paper is well written, with a clear, fluent and concise language and a well-organised structure. I think that this new method based on dynamical systems metrics can provide new insights into understanding the mechanisms behind compound events. Hence, my assessment of the manuscript is overall positive. However, I have to point out some deficiencies, which need to be fixed before publication: 1. The computation of $\alpha$ and especially of $\theta\hat{}(-1)$ is not described clearly and precisely, and I think it should not be substituted by merely a reference to another publication. The manuscript should contain the basic equations for the two main indicators (at least in the supplement) since these represent the core of the whole analysis. To assure the reproducibility of the results a precise description of the computational steps for $\alpha$ and $\theta\hat{}(-1)$ is required. 2. Approaches based on dynamical systems and extreme value theory are developed under certain assumptions, which are usually not entirely fulfilled in case of applications to geophysical data. These assumptions, together with their possible consequences to the results of the analysis, are not mentioned in the manuscript. Furthermore, the manuscript lacks a critical discussion related to the advantages and disadvantages of the applied method. The authors should discuss these very important points in the paper. 3. In the conclusion, the authors write that their results are in correspondence with previous studies. However, they do not point out clearly enough

the scientific gain based on this new work. What do we learn here we have not known before? This should be discussed thoroughly in the paper.

I would also welcome some comments about choosing $\alpha$=90th quantile as threshold for defining compound dynamical extremes. How robust are the obtained results against changes of this threshold? It would be also interesting to know what the authors think about the effect of the horizontal grid resolution on $\alpha$ and $\theta\hat{}(-1)$. A more detailed discussion of the spatial patterns of $\alpha$ and their possible connection to the atmospheric circulation would increase the quality of the paper as well.

SPECIFIC COMMENTS

It is difficult to compare the results for the different reanalysis products, because of the different axis or colour scale limits. For example, Fig. 4 – S7 / S9, Fig. 5 – S11 / S12, Fig. 6 – S13 / S14.

Fig.1(c) and 2(b): There seems to be no trend in $\theta\_(Tmax,P)\hat{}(-1)$ and in $\theta\_P\hat{}(-1)$ after 1995.

TECHNICAL CORRECTIONS

P4-L103: . . . higher (lower) than those observed. . .

FIG1: It is hard to see the difference between the two red lines.

P2-L7 Suppl.Data: "Tmax, Tmin and P" instead of "Tmax, Tmin and TP".

FIGS2 caption: "Tmin and P" instead of "Tmin and TP".

―――――――――――――――――――

---

## Author Comment (AC1) · 19 Jun 2020

**Response to Referee 1**

**We thank the referee for taking the time to review our manuscript. Please, find below your comments (CX) and our answers (AX), the latter highlighted in red.**

**Update:**

In the submitted paper there was an error on how we computed ERA5 and ERA5 ensemble total precipitation, ERA5 convective precipitation, ERA5 large-scale precipitation and ERA5 snowfall. The error was that we computed the daily total amount by simply summing 6-hourly time-steps, instead of summing all the 1-hourly time-steps for ERA5 and 3-hourly steps for ERA5 ensemble. Therefore, we rerun all the scripts and updated all the figures with the correct datasets. The procedure used to compute the new precipitation datasets was the following: i) shift back by 1 hour (3 hours for ERA5 ensemble) the time-steps; ii) sum the 24 (8 for ERA5 ensemble) values at daily resolution. Our results did not change significantly compared to the submitted paper, except for convective and large-scale precipitation (Figure S8 in the submitted paper) since now both datasets show statistically significant results (see new Figure S8_rev below). Therefore, we removed the old Figure S8 but computed the same for convective available potential energy (CAPE, $JKg^{-1}$). Results for CAPE (see Figure S8_new) show significant positive anomalies over the Alps during JJA and therefore we link the wet anomalies (Figure 4e) with localised convective P events. In the revised paper we replaced the old Figure S8 with Figure S8_new.

[Figure]

*Figure S8_rev - As Figure 4e but for daily anomaly means of (a) large-scale total precipitation (mm) and (b) convective total precipitation (mm).*

[Figure]

*Figure S8_new - As Figure 4e but for daily anomaly means of convective available potential energy (CAPE, JKg^-1).*

The authors investigate compound hot-dry and wet-cold events over the Mediterranean basin, employing a novel method based on dynamical systems theory. They use different reanalysis products and find a tendency towards an increasing coupling between temperature and precipitation over 1979-2018. The paper is well written and pleasant to read. I find the use/introduction of this approach interesting given that it offers a novel perspective for studying compound events. New approaches are always welcome as they can challenge or confirm previous findings. I recommend publishing the paper, but also to address some comments that follow. All of the comments are, of course, meant be constructive.

Thank you, please find our answers below.

**C1:** A main comment I have is about the definition of hot&dry and wet&cold conditions. I understand that they are defined based on positive/negative seasonal anomalies of precipitation and temperature. Although also non-extreme values of the contributing variables can lead to extreme impacts, the employed anomalies may be be particularly small. Given that the authors link the study directly to compound events and associated risks in the Mediterranean area, I think that some considerations are required. The authors could repeat some of the analyses based on higher anomalies (see comment below). Alternatively, I would recommend modifying the text in several parts, including abstract (e.g., line 7) and title, to avoid giving the impression of referring to, e.g., hot (and therefore extreme) events. In general, when possible some more physical interpretation would be welcome to guide the reader.

**A1:** We agree that within the text there is too much emphasis on extreme events, while we just considered compound anomalies. As suggested, we modified L7 and amended "hot-dry"

with "warm-dry" (title included) in the revised paper. Regarding physical interpretation of our results, we identified and described within the text two main physical processes: i) thermodynamic changes driving the increasing alpha and persistence trends during summer; and ii) the formation of a Cyprus-low over the Eastern MED during DJF. In addition to this, in the revised paper (Section 4.2) we added the physical interpretation of wet P anomalies during JJA over the Alps, which we now link to convective P events (see **Update** at the start of this file and A21). We also mention (Section 5)  that JJA SLP patterns may be a combination of different weather regimes (see also A20).

**Specific comments**

**C2:** L40, The paper from Manning et al. would definitely fit here (see also comment later): Manning C, Widmann M, Bevacqua E, Van Loon AF, Maraun D, Vrac M. Increased probability of compound long-duration dry and hot events in Europe during summer (1950– 2013). Environmental Research Letters. 2019 Aug 29;14(9):094006.

**A2:** The study is indeed relevant and a reference to it has been added in Section 1 of the revised paper.

**C3:** L42, I would strongly suggest adding 2-3 sentences in this paragraph. You could explain, via examples and references, why wet-cold and dry-hot events can lead to impacts (e.g., wildfire, vegetation issues etc), i.e. why they are important.

**A3:** As suggested, two sentences have been added in Section 1 of the revised paper highlighting hot-dry and cold-wet impacts.

**C4:** From line 61 onward. Overall, the explanation is very easy to understand. However, would it be possible to add 1-2 equations to guide some type of readers?

**A4:** We added the derivation of the metrics in Section 2.1 of the revised paper.

**C5:** L68, Does the persistence depend on any used threshold to define the close range dx (intorno) around Zeta_x? Or is it propriety of the system in Zeta_x that you somehow obtain based on some limits? Please, clarify.

**A5:** Yes, the persistence (and co-persistence) metric quantifies the mean residence time of the recurrences around a given state of interest Zeta_x (Zeta). Therefore, the persistence - as well as others dynamical indicators introduced in the manuscript - depends on the threshold (q) used to compute the recurrences. Such threshold refers to the size of the radius of the hyper-ball centred on Zeta_x. The higher the threshold q, the smaller the radius of the hyper-ball. In Section 2.1 of the revised paper we clarified this. The dependence of the results on the specific threshold used is discussed in several articles in the preceding literature (Faranda et al. 2011 J Stat Phys, Lucarini et al. 2014 J Stat Phys, Faranda et al 2017 Sci. Rep, Faranda et al 2019 Nature Comm). In this and those studies q=0.98 value was used. This value has provided good estimates of the dynamical indicators for two

reasons: on one hand it is high enough to ensure to select only genuine recurrences of Zeta, on the other it ensures a sufficient large sample of recurrences to perform statistical fits. In previous studies little sensitivity of the results to the threshold is found in the range $0.95 < q < 0.99$.

**C6:** L72, Also for the co-recurrence ratio: is this obtained based on (empirical) counting of the states and therefore it depends on the values dx and dy used to the define the close range around Zeta_x and Zeta_y? I agree that a full description of the theory should not be given but, in my opinion, a few sentences to guide the reader are needed here.

**A6:** Yes, the co-recurrence ratio is quantified by considering the joint number of recurrences between variable_x and variable_y. As for persistence, to compute recurrences we make use of thresholds for both variable_x and variable_y. Note that the thresholds for x and y are not the same, as we decided to used a fix a quantile of the recurrences for each variable. The same set of thresholds is used to compute persistences. In Section 2.1 of the revised paper we provide the derivation of the co-recurrence ratio.

**C7:** "Compound dynamical extremes", would it be better to use compound dynamical events? "Extremes" might be misleading.

**A7:** We appreciate the suggestion, however we would like to keep "compound dynamical extremes", since both recurrences and high-alpha values are computed by applying a high-quantile threshold to the time-series, and indeed their calculation issues directly from extreme value theory.

**C8:** L92, anomalies relative to JJA means

**A8:** We amended the text as suggested in the revised manuscript for both JJA and DJF definitions (Section 2.2).

**C9:** L96, Could you please explain why these slopes are preferred to usual linear regression. For example, would the trend in Fig. 1c-d be non-significant with a linear regression? Please, discuss this.

**A9:** We make use of the Mann-Kendall test to compute slopes and p-values because of its robustness to outliers and suitability for non-normal distributions. However, we tested some of our trends with linear regressions and confirm that both slopes and p-values reflect the ones computed using the Mann-Kendall test. Indeed, for JJA alpha trend (i) ERA5 (Figure 1a), (ii) ERA-Interim and (iii) ERA5 ensemble (Figure S1a) we find the following slopes and p-values with linear regressions: (i) $p < 0.01$ slope=0.001; (ii) $p < 0.01$ slope=0.0008; and (iii) $p < 0.01$ slope=0.0008.

**C10:** L110, Fig. 1a, Are you computing alpha for every JJA day and then computing the yearly average? Make this clearer, please (in the caption is not fully clear in my opinion).

**A10:** Yes, alpha (and persistence) has been computed first for the entire period 1979-2018, then filtered by JJA days and eventually JJA yearly means were computed. We amended the caption of Fig. 1a in the revised paper.

**C11:** L110, Guiding the reader to see what is happening in the (T,P) space would help here. For example, in summer, would you expect to find a similar trend using Tmin and P (or, in winter, Tmax and P)?

**A11:** Yes, similar trends are observed in winter when computing alpha with Tmax. Indeed, in a very early stage of the analysis we computed alpha for Tmin and P during JJA and found similar trends as for Tmax and P (see Figures R_1 and R_2). However, we specifically selected Tmax in summer and Tmin in winter to reflect our desired compound events (i.e. warm-dry and cold-wet). Such similarity between trends may be due to temperature, which, in both Tmax and Tmin cases is continuing increasing over the MED. We expanded the relevant sentence in the revised paper (Section 3).

[Figure]

*Figure R_1 - Alpha JJA trends computed using Tmin (instead of Tmax) and P for ERA5 reanalysis. Compare the figure with Figure 1a in the main paper.*

[Figure]

*Figure R_2 - Alpha JJA trends computed using Tmin (instead of Tmax) and P for ERA-Interim (blue) and ERA5 ensemble (black) reanalyses. Compare the figure with Figure S1a in the main paper.*

**C12:** L112, Would in any way carrying out the analysis after detrended the time series of the temperature help to better understand the physical drivers of the trends?

**A12:** Thanks for the comment. We opt for not showing such type of analysis since the detrending distorts the dynamical system's recurrences and would not satisfy the statistical assumptions underlying our approach. What we instead did was to explain the summer trends by plotting alpha and persistence ranked in ascending order by average Tmax over the 1979-2018 period.

**C13:** L 120, is there a correlation between co-persistence and alpha?

**A13:** We confirm that the correlation between JJA ERA5 alpha and persistence is positive (rho=0.75, Spearman's test) and significant (p<0.001). We mentioned this in Section 3 of the revised paper.

**C14:** L121, how are, in this regional case, hot and dry days subsampled?

**A14:** Here, the persistence trends have been computed *for each grid-point* and then averaged by considering only persistence daily values recorded during compound warm-dry events, instead of using the full time-series. We clarified the sentence also according to C2 by Referee 2 (Section 3).

**C15:** Section 4.1, Do also the univariate persistences show a similar seasonality? If so, is it possible to interpret this in relation to the seasonality in alpha?

**A15:** Thanks for the comment. We checked the seasonal trends of univariate persistences (Figures R_3-R_4) and co-persistence (Figure R_5) extremes (>90th quantile) for all three reanalysis products by making use of total precipitation (mm) and maximum temperature (K) data. We note that none of these plots agree with Figure 3a, and hence we do not find a marked and distinct peak in persistence extremes during July and August months, except for precipitation (Figure R_3), which may be linked to the prevailing dry conditions over the region. We added a sentence referring to this in Section 4.1 of the revised paper.

[Figure]

*Figure R_3 - Univariate total precipitation (mm) persistence extremes (>90th quantile) computed for the three reanalysis products within the 1979-2018 period.*

[Figure]

*Figure R_4 - Univariate maximum temperature (K) persistence extremes (>90th quantile) computed for the three reanalysis products within the 1979-2018 period.*

[Figure]

*Figure R_5 - Co-persistence extremes (>90th quantile) computed for the three reanalysis products within the 1979-2018 period by making use of total precipitation (mm) and maximum temperature (K) data.*

**C16:** Section 4.2 and the following sections. The authors could consider whether moving these results before the trends would help or not. Being aware of what alpha depicts from a physical point of view may help to interpret the trends more easily.

**A16:** Thank you for the suggestion. We would like to keep the structure of the paper as it is now. The reason is that Section 3 shows trends only for JJA, whereas Sections 4.2-4.4 show both JJA and DJF and therefore the overall flow of the paper may be disrupted if we were to move these sections before Section 3.

L141 (paragraph)

**C17:** 1) In general, a discussion on why one would expect to capture anomalies would be important to help the reader.

**A17:** At the end of Section 5 we indeed discuss the link between CDEs and compound events. However, to strengthen our point a new sentence has been added at the end of Section 5 of the revised paper, stating that "a link between CDEs and compound events is expected because in both cases the data reflect anomalous (or high-coupled) conditions for the same atmospheric variables".

**C18:** 2) I understand that the anomalies are computed relative to the seasonal mean, please specify it.

**A18:** In Section 2.2 of the revised paper we made clear that anomalies have been computed from seasonal means.

**C19:** 3) To strengthen the conclusions, you could highlight that small anomalies are expected over the sea due to water inertia, even during heatwaves.

**A19:** Thanks for raising a good point. A sentence specifying this has been added in Section 5 of the revised paper.

**C20:** 4) Overall, the SLP field picked up by high alpha values in winter appears associated with a more defined atmospheric configuration compared to summer. Is it possible that you pick up different weather circulations within the subsample of extreme alpha in summer? - I see you have stippling in Fig. 4. How large is the average anomaly during days with large alpha compared to the standard deviation*? *computed based on the daily anomaly data. - How do these maps change when using, e.g., the 95th percentile to define extreme alpha?

**A20:** Yes, during winter the SLP synoptic patterns clearly reflect a well documented phenomenon over the eastern MED (i.e. the Cyprus-low), whereas the SLP patterns in summer do not match any large-configuration we know of. Therefore, it may be indeed possible that alpha extremes in summer capture a different set of weather regimes. Before submitting the manuscript, we tried to investigate the matter with a Self Organising Maps (SOMs) analysis. However, the SOMs results did not point to any plausible physical mechanism, so that we do not show them. Nonetheless, we now specify in the revised manuscript (Section 5) that results obtained for summer might show a composition of different weather circulations that we are unable to isolate clearly. Please find below in Figure R_6 the ratio between anomaly means and SDs corresponding to Figure 4. We note that the JJA season displays similar ratio values, albeit with different spatial patterns, when compared to DJF.

[Figure]

*Figure R_6 - As Figure 4 in the main manuscript but for the ratio between anomaly means and their SDs.*

Figures R_7-R_9 below, reproduce Figures 4, S7 and S9 of the submitted paper but with anomaly means computed from alpha extremes >95th quantile. In general, the figures are in agreement with the ones in the submitted paper for all variables (i.e. SLP, Tmax, Tmin and P), seasons and reanalysis products. The only difference are the anomaly values which in Figures R_7-R_9 are larger compared to Figures 4, S7 and S9 (see colorbars), but this is somehow expected since we now compute anomaly means based on CDEs derived from a higher threshold (95th quantile). In Figure R_10 we also show the ratio between anomalies and SDs of Figure R_7. As you see the ratio spatial patterns between Figure R_6 (alpha>90th quantile) and Figure R_10 (alpha>95th quantile) are in agreement, although the ones in Figure R_6 are weaker. This suggests that our main results are not overly sensitive to the alpha threshold used. We added new sentences mentioning this in Section 2.1 and Section 4.2 of the revised paper (see also A4 of Referee 3).

[Figure]

*Figure R_7 - As Figure 4 but for alpha extremes > 95th quantile.*

[Figure]

*Figure R_8 - As Figure S7 but for alpha extremes > 95th quantile.*

[Figure]

*Figure R_9 - As Figure S9 but for alpha extremes > 95th quantile.*

[Figure]

*Figure R_10 - As Figure R_6 but for ratios computed from alpha extremes >95th quantile.*

**C21:** 5) L146, The latter correspond…: Personally, I would rephrase given that although the significant anomaly is all associated with the large-scale component of the precipitation, also the convective part is relevant. The reasoning in the next sentence would still work, maybe saying these anomalies are \*mainly\* linked to SLP.

**A21:** We removed the sentence in L146-147 along with Figure S8 (see **Update** at the start of this file). In addition, we computed ERA5 JJA anomaly means of convective available potential energy (CAPE, JKg^-1) occurring during alpha extremes and found positive and significant CAPE anomaly means over the Alps (Figure S8_new). Therefore, we conclude that the JJA wet P anomalies observed over the Alps (Figure 4e) are driven by localised convective P events. In the revised paper we replaced the old Figure S8 with Figure S8_new and added a sentence specifying the findings in Section 4.2.

[Figure]

*Figure S8_new - As Figure 4e but for daily anomaly means of convective available potential energy (CAPE, JKg^-1).*

**C22:** L158, The sentence is correct. But it suggests that these events can occur everywhere over the analysed domain, while, especially the P anomalies, suggest that this is the case mainly over the eastern domain and along the Italian areas exposed to cold-air advection from northern Europe.

**A22:** We clarified that CDEs match compound cold-wet events "over the eastern MED" in the revised paper (Section 4.2).

**C23:** L160, I understand that hot-dry and cold-wet events are defined based on positive/negative anomalies from the seasonal average. Would the main conclusions be similar if using larger anomalies to define, hot/cold and wet/dry conditions? For example, one could use +/-2 standard deviations from 0 to define larger anomalies.

**A23:** Thank you for the comment. In the next revision (after Editor's comments) we will try to replicate some of our figures with *extreme* anomalies as you suggested.

**C24:** L167, Could you relate to the numbers above, i.e., does this also imply that also the values 84% and 77% are significantly large?

**A24:** These percentage values reflect the cumulative distribution functions (CDFs) of anomaly means of interest for all grid-points (Figure 5a-b). In other words, 84% of Tmax anomaly means are warm (>0 K) and 77% of P anomaly means are dry (<0 mm). The same (but for cold-wet events) applies to DJF percentage values (Figure 5c-d). Here we checked the statistical significance between distributions of anomaly means during alpha and non-alpha extremes (Section 2.3 and 4.3) and found statistically significant p-values, therefore we intuitively expect that the percentage values are significant too. Note that in the

revised manuscript the percentage values changed slightly because we made use of the correct ERA5 total precipitation dataset (see **Update** at the start of this file).

**C25:** L162, How are these numbers computed: Are all CDE days and grid points pulled together? Please, clarify.

**A25:** Please see comment C24 above.

**C26:** L168, consistently with Fig. 4f, the distribution of the precipitation is peaked around zero in fig. 5d. I am wondering whether (maybe for future work) the authors would see any added value in focussing some of these statistics only over the eastern part of the domain, where the framework is able to better capture anomalous conditions. If so, this could be discussed.

**A26:** Thanks for the comment and suggestion. It would be definitely interesting to compute the dynamical systems metrics only for the eastern MED subregion, since at least in DJF it is the one showing a clear and well documented synoptic pattern. At present, works by Hochman et al. partly address your question, but for future work we see our interests moving towards a larger-scale dynamical systems analysis instead of focussing on smaller regions. We specified this at the end of Section 5 in the revised paper.

**C27:** Section 4.4, It is a bit difficult to read the values in fig. 6 given that the palette has continuous values. Aren't these values depending on the percentiles (here 90th) used to define the CDE events? Therefore, the reader should be helped to interpret these numbers. They should be compared to what expected under a certain null hypothesis. For example, one could easily compute the probability of getting concurrent CDE and hot&dry days assuming that the CDE events are randomly distributed during the year (if this is a reasonable assumption).

**A27:** Thanks for your comment. As suggested, in the next revision (after Editor's comments) we will try to compute the statistical significance for Figures 6, S13-S14.

**C28:** Discussion: Could you add 1-2 sentences about the expected sensitivity of the results from the size of the analysed domain? This is relevant for the reader...

**A28:** These have been added in the revised paper.

**C29:** L190, Do you think that re-computing the trends in the two metrics obtained based on maps of (1) land surface only and of (2) sea surface only could somehow allow for speculating more safely about this? Or, more in general, could this allow for disentangling a higher signal of the increasing coupling on land?

**A29:** Yes, computing the dynamical systems metrics based only on land-surface (and/or sea-surface) data may help in providing more understanding about the physical processes at play during summer. Temperature-precipitation coupling may change significantly between land and sea, due to the very different thermal inertiae of the underlying surfaces, since in

the former many components of the earth's surface affect the coupling (e.g. vegetation, orography, built environment and freshwater systems), whereas in the latter the Clausius-Clapeyron relation is followed with no (or little) disturbances. In the next revision (after Editor's comments) we will try to replicate Figure 1 for land- and sea-only data.

**C30:** I understand that the trends you found in the regional coupling can have different meaning depending on the areas of the domain so I understand that they should be interpreted bearing this in mind. Could you discuss more explicitly on this from an impact perspective?

**A30:** A sentence with a practical examples has been added in the revised paper.

**C31:** Around L205, I would suggest to interpret and discuss the results also in relation to results found by Manning et al. (paper cited above).

**A31:** A new sentence discussing the results of Manning et al. (2019) has been added in the revised paper.

**C32:** L215, see my main comment about the definition of hot days.

**A32:** We amended 'hot' with 'warm' throughout the text and in the title of the revised manuscript.

**C33:** The authors could consider expanding the discussion, very briefly, to highlights the potential benefits of their approach for the part of the compound event community that is focussing on impact assessments.

**A33:** We expanded the discussion in the revised paper. See also comment A30 above.

**More technical comments**

We thank the Referee for spotting these typos and small errors which had escaped us.

**C34:** L10, discussing first the winter results may allow avoiding some repetitions.

**A34:** Thank you. The abstract has been amended as suggested.

**C35:** L17, add space

**A35:** Done.

**C36:** L19, I would probably start talking of atmospheric circulation changes which are more intuitive than "dynamical changes" for a non-expert reader.

**A36:** The word has been amended as suggested.

**C37:** L20, may expect based on?

**A37:** "as a consequence of global warming". We amended the sentence.

**C38:** L27, an increase in "daily" or "episodic" precipitation extremes?

**A38:** To be more generic, we amended the sentence with "heavy precipitation events" in the revised paper.

**C39:** L56, "[...]. The metric theta^{-1}..."

**A39:** This has been added in the sentence.

**C40:** L79, each daily timestep

**A40:** "daily" added in the sentence.

**C41:** L80, with -> characterized by…;

**A41:** Amended.

**C42:** L81, could be written slightly better

**A42:** Thanks. The sentence has been rephrased.

**C43:** Fig.1, caption, L4, 5 year centered moving [...]

**A43:** "centered" added in the caption.

**C44:** L200, Please, check that it is ok. I assume you did not talk of Tmax *and negative P* anomalies on purpose.

**A44:** In the revised paper we clarified the sentence by adding "and negative P anomalies" when referring to JJA.

**C45:** L 211, it is probably more correct to write "concurrent cold spells and heavy…"

**A45:** "concurrent" has been added in the sentence.

**C46:** L212, "e.g" in the parenthesis

**A46:** Thanks. This has been corrected.

**C47:** L212, "accordance" a better word? It might not look in accordance with the decrease found in the other studies.

**A47:** We amended the word with "in line" to reduce the emphasis of the sentence.

---

## Author Comment (AC2) · 19 Jun 2020

**Response to Referee 2**

**We thank the referee for taking the time to review our manuscript. Please, find below your comments (CX) and our answers (AX), the latter highlighted in red.**

**Update:**
After checking carefully the ERA5 datasets used in the analysis we found an error when computing total precipitation. We therefore corrected the error and created new figures. No significant changes, compared to the submitted version of the paper are observed, except for Figure S8 which has now been removed and replaced with Figure S8_new. Please see the **Update** description in "Response to Referee 1" for more details.

[Figure]

*Figure S8_new - As Figure 4e but for daily anomaly means of convective available potential energy (CAPE, JKg^-1).*

In this paper concurrent recurrence of temperature and precipitation patterns over the Mediterranean are quantified and changes over time highlighted. The paper is well written and the results are highly interesting and relevant. There are some methodological points that need further explanations for non-experts (explained in more detail under major points) and the colorbars of the map figures need to be improved. After these clarifications the paper is ready for publication. Olivia Martius

Thank you, please find our answers below.

Major points:

**C1:** 1) Consider adapting the title because the term "dynamical extremes" may not be widely known.

**A1:** Based also on comments from Referee 1, we amended the title to "Compound Warm-Dry and Cold-Wet Events Over the Mediterranean".

2) I have some basic methodological questions that I did not yet fully understand and that might be confusing for other readers as well.

**C2:** a. Co-occurrence of dry and hot (cold and wet) conditions refers to the entire Mediterranean area. The dry conditions and the hot conditions must not necessarily happen at the same location in the Mediterranean area?

**A2:** Yes, warm-dry (cold-wet) conditions are computed singularly at grid-point-level (based on alpha extremes) and therefore if a grid-point in a given day is warm it does not necessarily imply that it is also dry. The *average* of these patterns are shown at grid-point levels in Figure 4c-f. We expanded a sentence in Section 2.2 specifying this.

**C3:** b. You state that you get an alpha and theta value for each time step of the data, but it is unclear how long the time period is that you use to determine these parameters, you only mention a "relatively long time period" in the beginning. Is it a moving window of a number of days? Are then successive alpha and theta values very similar?

**A3:** We computed alpha and theta for the entire 1979-2018 period at daily timesteps. We mentioned "relatively long time period" when providing the general description and derivation of the dynamical systems metrics. However in the revised paper we specified the 1979-2018 period within this sentence. Moreover, at the end of Sections 2.1 and 2.2 we also clearly stated that alpha and theta have been computed at daily timsteps for the full 1979-2018 period. Within this study we did not use a moving window nor aggregated data over *n* days.

**C4:** c. Is it correct that compound dynamical extremes are "extreme" in the coupling but not necessarily in p and T? Do these CDEs then point to weather situations that are dominated by the synoptic-scale flow rather than by local convective systems?

**A4:** Yes, the alpha metric reflects the strength of joint recurrences in the phase-space, which are defined with a high-quantile threshold (see updated Section 2.1 in the revised paper). Then, based on daily alpha values (1979-2018), we apply a further threshold (in our case >90th quantile) and define alpha observations > 90th quantile CDEs. For a test of the sensitivity of our results to this choice of threshold see our reply to C20 from Reviewer 1 above. So CDEs are indeed extremes in the coupling by definition. Lastly, we compute composite maps of SLP, Tmax (Tmin) and P based on CDE days and prove that CDEs correspond to warm-dry and cold-wet days over the MED, although these are not necessarily extremes. Such link naturally depends on the variables, seasons and region(s) under study.

In the submitted manuscript we computed composite maps of large-scale and convective P during CDEs in JJA (Figure S8) and found that large-scale P was having a significant effect compared to convective P. However, after updating Figure S8 with large-scale and convective P calculated correctly (see **Update** at the start of this file) we found that both

fields show significant anomalies (Figure S8_rev). Therefore, we explored further the matter by making use of convective available potential energy (CAPE, JKg^-1) and found that its positive anomalies over the Alps during JJA are statistically significant (Figure S8_new). Therefore, we concluded that JJA wet anomalies during alpha extremes over the Alps are driven by localised convective P events.

[Figure]

*Figure S8_rev - As Figure 4e but for daily anomaly means of (a) large-scale total precipitation (mm) and (b) convective total precipitation (mm).*

**C5:** 3) Please use colorbars with discrete colors for all map figures. For example, when I am interested in the SLP anomaly over Italy in Figure 4a it is very difficult to link the discrete colors on the map to the continuous colors in the colorbar.

**A5:** Thank you for your comment. In the next revision (after Editor's comments) we will try to re-do all the maps with discrete colorbars.

**C6:** 4) The interpretation of Figure 4 is difficult without a reference to the climatology. It would be good to indicate how the anomalies relate to the climatology (e.g. % of mean) and if feasible to the variability (e.g., STD, inter quartile range). For example, precipitation rates in the Alps are higher in the climatology. So maybe in Figure 4e the anomalies over the Alps are small from a climatological and variability perspective and the dry anomalies in other areas are large.

**A6:** Thank you for your comment. The statistical test performed in Figure 4 (one-tailed Mann-Whitney test) is an ordinal test which has been applied singularly to each grid-point. As such, it provides a statistical comparison of the values between the two sets of data regardless of their magnitude. With respect to the variability, we computed the standard deviations (SDs) of the anomaly means reported in Figure 4 (Figure R_1). From Figure R_1 one can see that: i) SLP SDs are larger over the northern and central MED; ii) temperature SDs are larger over land compared to the sea; and iii) precipitation SDs are larger where the higher anomaly values are reported (i.e. the Alps in JJA and south-eastern MED in DJF). See also answer A20 to Referee 1. A sentence describing the SD patterns has been added in Section 4.2 of the revised paper.

[Figure]

*Figure R_1 - Standard deviations (SDs) computed for the anomaly means of Figure 4 in the main text.*

**C7:** This would also clarify a follow up question namely can your method capture dry and wet extremes at the same time? The discussion of Figure 4d focuses on the wet anomaly over the Alps but the subsequent discussion of the histograms points to the prevalent dry signal. This is confusing for the reader.

**A7:** Yes, our method can also capture dry and wet extremes co-occurring in different parts of the geographical domain under study. Indeed, our method captures the dependence structure of two atmospheric variables, independently of their values (e.g. wet or dry) and spatial configuration. We then focus our analysis on a specific combination of anomalies based on their prevalence throughout the analysis domain. We clarified this in Section 2.1 of the revised paper.

Minor points:

**C8:** 1) L16: Could you provide an example for the dynamical changes.

**A8:** We added northward shift and intensification of the storm track as examples of atmospheric circulation changes as discussed in Hoerling et al. (2012).

**C9:** 2) L28: Clarify what you mean by similar changes.

**A9:** The sentence has been rephrased.

**C10:** 3) L44: Define large-scale precipitation, here I assume it is the model variable.

**A10:** Thanks. In this case "precipitation" was meant to be generic. The word "large-scale" has been removed.

**C11:** 4) L47: Suggest to shift a couple of sentences from the next section up here to briefly explain what dynamical systems are. A dynamical meteorologist might think of weather systems at this point of the paper rather than system dynamics.

**A11:** Two sentences have been added as suggested.

**C12:** 5) L92: How do you define anomalies? Are these to the 30-year (seasonally varying?) mean?

**A12:** Anomalies are defined based on the JJA (DJF) seasonal means. We specified this in Section 2.2 as also suggested by C8 of Referee 1.

**C13:** 6) L92: But you also use large-scale vs. convective precipitation? Could you add the information that the precipitation is a forecast field and not assimilated.

**A13:** Yes, in the first submitted file we also use large-scale and convective precipitation, please see **Update** at the start of this file. We added in Section 2.2 and supplementary material that precipitation is a forecasted field while temperature, sea-level pressure and CAPE are instantaneous fields.

**C14:** 7) L121: what do you mean exactly be "restricting the analysis. . ."?

**A14:** By "restricting the analysis" we meant that we computed the co-persistence trend by using only warm-dry days instead of the full dataset. Specifically, the persistence trends have been here computed *for each grid-point*, and then averaged, by considering only persistence daily values recorded during compound warm-dry events, instead of using the full time-series. The sentence has been amended.

**C15:** 8) L141: Please define how you calculate the anomalies, wrt to a seasonal mean?

**A15:** Yes, SLP anomalies have been computed with respect to JJA (DJF) means. We made this clear in Section 2.2 of the revised paper.

**C16:** 9) L141: Linking the precipitation anomalies with the SLP anomalies should be formulated in the form of a hypothesis.

**A16:** The sentence has been rephrased as suggested in the revised paper.

**C17:** 10) L147: Please add supporting references. For heavy precipitation along the western Alpine south-side the low pressure system is typically located over the Gulf of Genoa. Also these types of low pressure systems are associated with cold fronts and colder surface temperature. However, the temperature pattern shows high temperatures in this area. An alternative interpretation is that the low pressure system over the Balkans might correspond to a heat low. It is unclear for me which aspects of the SLP distribution are related to the precipitation field and which ones to the temperature field. Can you separate this? This might also link back to major point 4.

**A17:** We agree that for the western Alpine south-side heavy precipitations are mostly associated with Genoa Low pressure systems. Following the study by Trigo, Isabel F., Grant R. Bigg, and Trevor D. Davies. "Climatology of cyclogenesis mechanisms in the Mediterranean." *Monthly Weather Review* 130.3 (2002): 549-569, we note however that the large scale circulation pattern induced by Genoa Low has a less pronounced anomaly structure over the MED than the Cyprus low, as the main cyclonic structure in these events is still typically on the Northern side of the Alps. Moreover, we tend to disagree on the nature of cold fronts and colder SST associated with Genoa Low. They mostly produce occluded or warm fronts as the results of the advection of Scirocco (SE) or Libeccio (SW) winds from Africa. These winds, charged with the humidity of the Mediterranean sea and impacting with the Apennines and with the Southern Alpine mountain ranges, produce heavy precipitation with Stau effects (Faccini, F., et al. "Geohydrological hazards and urban development in the Mediterranean area: an example from Genoa (Liguria, Italy)." *Natural Hazards & Earth System Sciences* 15.12 (2015)). In the same paper and references therein it is well explained why Genoa lows extreme precipitations are rather associated with warm fronts than cold fronts.

However, in our specific case (see Figure R_2 below) we did not find the signature of the Genoa low during JJA and after checking JJA CAPE anomaly means observed during alpha extremes we conclude that wet P anomalies over the Alps in JJA are driven by local convective events. We therefore: i) removed from L148-149 of the submitted paper "and in particular to the advection of moist Mediterranean air masses inland towards the Alpine region." ; and ii) we specified the link between wet P anomalies and convective events in Section 4.2 of the revised paper.

[Figure]

*Figure R_2 - New Figure 4 of the revised paper with total precipitation computed correctly (see **Update** at the start of this file).*

**C18:** 11) L151: How does the cold air from northern Europe cross the Alps into Italy? This statement does not have any supporting analyses in the paper. Please either refer to the literature, show the trajectories or remove the statement.

**A18:** In the revised paper we amended the sentence as follows: "In DJF we observe an east-west dipole in SLP over the MED, that favours cold-air advection from northern Europe to the the Balkans and the eastern MED, leading to significant negative Tmin anomalies in these regions. Negative and significant Tmin anomalies are also observed over the Italian peninsula, central and western MED."

**C19:** 12) L162: how do you define dry?

**A19:** Dry anomalies are negative (<0mm) precipitation anomaly values computed relative to the seasonal JJA means. We improved the definition of anomalies in Section 2.2 of the revised paper.

**C20:** 13) Please add units to Figure 5.

**A20:** Done.

**C21:** 14) L175: It is unclear to me how you compute these maps since I understood the measures to be linked to one pattern over the entire Mediterranean. Please expand your explanations.

**A21:** Here we link the compound events for each gridpoint to CDEs (i.e. one pattern or time-series for the entire MED). The procedure to compute the maps is the following: i) for each grid-point in Figure 6 we identify the days (or dates) reporting compound events *and* CDEs, ii) we divide the total number of these days by the total number of CDEs and iii) we multiply the resulting number by 100 to obtain the % agreement value. A sentence has been added in the revised paper that clarifies the method.

**C22:** 15) L195: weaker anomalies -> can you be more specific?

**A22:** By 'weaker anomalies' we refer to the fact that P anomalies during JJA are mostly negative, yet not large enough to result in statistically significant values (see Figure 4e). We replaced 'weaker' with 'not significant' in the revised paper.

---

## Author Comment (AC3) · 19 Jun 2020

**Response to Referee 3**

**We thank the referee for taking the time to review our manuscript. Please, find below your comments (CX) and our answers (AX), the latter highlighted in red.**

**Update:**
After checking carefully the ERA5 datasets used in the analysis we found an error when computing total precipitation. We therefore corrected the error and created new figures. No significant changes, compared to the submitted version of the paper are observed, except for Figure S8 which has now been removed and replaced with Figure S8_new. Please see the **Update** description in "Response to Referee 1" for more details.

[Figure]

*Figure S8_new - As Figure 4e but for daily anomaly means of convective available potential energy (CAPE, JKg^-1).*

GENERAL COMMENTS

The authors analyse hot-dry as well as cold-wet dynamical extremes over the Mediterranean region in ERA reanalysis data sets over the period 1979-2018. They use a novel method based on dynamical systems metrics and extreme value theory to select and analyse so-called "compound dynamical extremes." The study is mainly based on two indicators, termed the co-recurrence ratio α and the co-persistence θ^(-1). They estimate these indicators for joint occurrences of daily maximum temperature and total precipitation, as well as of daily minimum temperature and total precipitation. They define the events with α>90th quantile of the whole α distribution as compound dynamical extremes. The authors find a positive trend in the co-recurrence and co-persistence of hot-dry events during summer (JJA), whereas no trend can be found in case of the co-recurrence of cold-wet events in winter (DJF). Thus, they conclude that long-term warming strengthens the coupling between temperature and precipitation, leading to more intense hot-dry compound events. They also

analyse spatial fields of sea level pressure, temperature and precipitation during compound dynamical extremes, as well as spatial maps of the co-recurrence ratio α.

The paper is well written, with a clear, fluent and concise language and a well organised structure. I think that this new method based on dynamical systems metrics can provide new insights into understanding the mechanisms behind compound events. Hence, my assessment of the manuscript is overall positive. However, I have to point out some deficiencies, which need to be fixed before publication:

Thank you, please find our answers below.

**C1:** 1. The computation of α and especially of θˆ(-1) is not described clearly and precisely, and I think it should not be substituted by merely a reference to another publication. The manuscript should contain the basic equations for the two main indicators (at least in the supplement) since these represent the core of the whole analysis. To assure the reproducibility of the results a precise description of the computational steps for α and θˆ(-1) is required.

**A1:** In the revised paper (Section 2.1) the complete derivation of θˆ(-1) and α has been added as suggested.

**C2:** 2. Approaches based on dynamical systems and extreme value theory are developed under certain assumptions, which are usually not entirely fulfilled in case of applications to geophysical data. These assumptions, together with their possible consequences to the results of the analysis, are not mentioned in the manuscript. Furthermore, the manuscript lacks a critical discussion related to the advantages and disadvantages of the applied method. The authors should discuss these very important points in the paper.

**A2:** We agree that the limitations and the hypothesis of the theory should be specified in the present study. In Section 2.1 of the revised paper we have therefore added: "The recurrence approach for the computation of the dynamical indicators is based on the following assumptions: 1) the existence of an underlying chaotic attractor for the dynamics (Freitas et al. 2010), 2) the quasi-stationarity of the dynamics: the method can handle dynamics where weak nonstationarities are present in the dynamics (see e.g. Faranda et al. 2019 Nature Comm). The method cannot be used when the nonstationarities lead to bifurcations of the system." As for the advantages: "with respect to statistical techniques, the dynamical indicators provide information on the underlying nature of the dynamics, i.e. the fact that co-recurrences are associated to high or low values of persistence and predictability connect co-recurrences to specific points of the phase space (unstable orbits, periodic points) (Faranda et al. 2020 Clim Dyn)."

**C3:** 3. In the conclusion, the authors write that their results are in correspondence with previous studies. However, they do not point out clearly enough the scientific gain based on this new work. What do we learn here we have not known before? This should be discussed thoroughly in the paper.

**A3:** Thank you for raising this point. In the Section 5 of the revised paper we added three main points highlighting the scientific gain obtained from our work. These are: i) the coupling between temperature and precipitation at large scale is driven by specific regions and processes (e.g. Cyprus-low) and therefore it does not always reflect the whole region under study; ii) the coupling results are highly sensitive even to non-extreme events, thus providing added value to conventional climatological analyses; and iii) our results provide information on specific factors that are driving the changes in the coupling of the variables being analysed (e.g. surface warming).

**C4:** I would also welcome some comments about choosing α=90th quantile as threshold for defining compound dynamical extremes. How robust are the obtained results against changes of this threshold? It would be also interesting to know what the authors think about the effect of the horizontal grid resolution on α and $\theta^{(-1)}$.

**A4:** In the initial stage of the research part of the analysis was performed with three α thresholds to test the sensitivity of the results: i) Q90th; ii) Q95th; and iii) Q99th. We found no significant differences when changing the α threshold and we make use of Q90th because a larger number of CDEs are available (compared to Q95th and Q99th), which can eventually provide more robust results when composited with Tmax (Tmin), P and SLP anomalies. We added two sentences referring to this in Section 2.1 of the revised paper.

Please, see below Figures R_1-R_3 which reproduce Figures 4, S7 and S9 in the submitted paper but for anomaly means computed from alpha extremes >95th quantile. As you can see, when changing the alpha threshold the synoptic patterns are kept and the only (expected) difference is the size of the anomaly means, which in the case of alpha>95th quantile are larger compared to alpha>90th quantile. We added a sentence specifying this in Section 4.2 of the revised paper (see also A20 of Referee 1).

[Figure]

*Figure R_1 - As Figure 4 but for alpha extremes > 95th quantile.*

[Figure]

*Figure R_2 - As Figure S7 but for alpha extremes > 95th quantile.*

[Figure]

*Figure R_3 - As Figure S9 but for alpha extremes > 95th quantile.*

Horizontal grid resolution does not change significantly the α and θˆ(-1) values. This is confirmed in our work by looking at ERA-Interim (0.75deg) and ERA5 ensemble (0.5deg) results in the Supplementary Material. Such datasets have a coarser horizontal resolution compared to ERA5 (0.25deg) but still their trends, composites and spatial compound maps are in agreement with the ERA5 ones. We highlight this point when presenting the results for ERA5 throughout the text and also added two sentences in the revised paper (Section 5).

**C5:** A more detailed discussion of the spatial patterns of α and their possible connection to the atmospheric circulation would increase the quality of the paper as well.

**A5:** In DJF we found that SLP patterns composited on CDEs show a Cyprus-low over the eastern MED. This has been mentioned and discussed in the submitted paper. During JJA, the SLP patterns do not point to any documented synoptic configuration. Before submission we verified whether this may have depended on the presence of multiple, distinct clusters of configurations by using Self-Organising Maps (SOMs). However, for JJA results we provide

a physical explanation of the α trends, which we found are driven by surface MED warming and prove that wet P anomalies are driven by convective P events (see **Update** at the start of this file). We also computed the standard deviations of the anomalies observed in Figure 4 (Figure R_4). See also answer A20 of Referee 1 and answer A6 of Referee 2.

[Figure]

*Figure R_4 - Standard deviations (SDs) computed for the anomaly means of Figure 4 in the main text.*

SPECIFIC COMMENTS

**C6:** It is difficult to compare the results for the different reanalysis products, because of the different axis or colour scale limits. For example, Fig. 4 – S7 / S9, Fig. 5 – S11 / S12, Fig. 6 – S13 / S14.

**A6:** In the revised paper we redid the axes of the colorbars for all the maps so that a comparison between the different reanalysis products can be made.

**C7:** Fig.1(c) and 2(b): There seems to be no trend in $\theta_{(Tmax,P)}^{-1}$ and in $\theta_P^{-1}$ after 1995.

**A7:** Thank you for the observation. In this work we assessed the full trends from 1979 to 2018 and would like to keep sub-trend analysis, which may be more closely linked to low-frequency modes of climate variability, for further work.

TECHNICAL CORRECTIONS

**C8:** P4-L103: . . . higher (lower) than those observed. . .

**A8:** Text amended.

**C9:** FIG1: It is hard to see the difference between the two red lines.

**A9:** In all the figures showing trends we amended the thin lines (5-year moving averages) with dashed lines.

**C10:** P2-L7 Suppl.Data: "Tmax, Tmin and P" instead of "Tmax, Tmin and TP".

**A10:** Text amended.

**C11:** FIGS2 caption: "Tmin and P" instead of "Tmin and TP".

**A11:** Text amended.

---

## Author Response (AR1)

**Response to Editor**

**We thank the editor for taking the time to review our manuscript. Please, find below your comments and our answers, the latter highlighted in red. Further below, you will find the remaining comments (CX) and our answers (AX) to Referees 1 and 2 which were not addressed in full before your first decision. Lastly, the revised manuscript with track changes highlighted can be found after the Referees responses.**

The authors find an increase in Tmax-P coupling based on alpha (section 3). How much (if anything at all) of this increases can be explained by an increase in temperature alone?

This is a very good point, which we had overlooked. We replicated Figure 1b for average MED P ranked *descendingly* (as a proxy of increased dryness) and found that the alpha trend is positive and statistically significant (see Figure R_1 below). Moreover, the correlation between the alpha values in Figure 1b and Figure R_1 is positive and statistically significant (rho=0.56, p-value<0.01). However, at this stage is difficult to discern between Tmax and P roles in driving the JJA alpha trend, since they may have a compound (Tmax *and* P) or a univariate (Tmax *or* P) effect on alpha. We will therefore keep this investigation for a further work, but added Figure R_1 in the Supplementary Material and mentioned what has been said above in Section 5 of the revised paper.

[Figure]

*Figure R_1 - As Figure 1b but for average P ranked descendingly.*

I would like to see a bit more discussion on this, in particular because you relate these finding to Zscheischler & Seneviratne (2017) in L 204. However, Zscheischler & Seneviratne found an increase in summer T-P coupling in CMIP5 after subtracting long-term trends. Hence, here the projected increase in coupling comes in addition to long-term climate change. Is your approach able to detect changes in coupling in a non-stationary climate?

We expanded the discussion following your first comment and now specify in the text that Zscheischler & Seneviratne (2017) find increased coupling without long-term trends, contrary to our analysis which is on raw data. Strictly speaking, our method is applicable to ergodic systems. In practice, it may be successfully applied to weakly non-stationary systems, as long as the non-stationarity is not so strong as to preclude the occurrence of recurrences of the system to previously visited states. From previous work by some of the authors (e.g. Rodrigues et al., 2018), we find that the historical climate fulfills the latter requirement.

Reference
*Rodrigues, D., M. C. Alvarez-Castro, G. Messori, P. Yiou, Y. Robin, and D. Faranda, 2018. Dynamical Properties of the North Atlantic Atmospheric Circulation in the Past 150 Years in CMIP5 Models and the 20CRv2c Reanalysis. J. Climate, **31**, 6097–6111*

**Remaining responses to Referee 1**

**We thank the referee for taking the time to review our manuscript. Please, find below your remaining comments (CX) and our answers (AX), the latter highlighted in red.**

**C23:** L160, I understand that hot-dry and cold-wet events are defined based on positive/negative anomalies from the seasonal average. Would the main conclusions be similar if using larger anomalies to define, hot/cold and wet/dry conditions? For example, one could use +/-2 standard deviations from 0 to define larger anomalies.

**A23:** Thank you for the comment. We re-computed Figure 4 by using anomalies > 90th and anomalies < 10th quantiles (Figure R_2). The results are in general agreement with Figure 4, except that in JJA the positive SLP anomalies are less in number. We mentioned this finding in the revised paper (Section 4.2) and added Figure R_2 in the Supplementary Material.

[Figure]

*Figure R_2 - As Figure 4 in the main text but for anomalies > 90th and < 10th quantile.*

**C27:** Section 4.4, It is a bit difficult to read the values in fig. 6 given that the palette has continuous values. Aren't these values depending on the percentiles (here 90th) used to define the CDE events? Therefore, the reader should be helped to interpret these numbers. They should be compared to what expected under a certain null hypothesis. For example, one could easily compute the probability of getting concurrent CDE and hot&dry days assuming that the CDE events are randomly distributed during the year (if this is a reasonable assumption).

**A27:** Thanks for your comment. We amended all the colorbars in all maps from continuous to discrete (see also A5 Referee 2). We also performed a statistical significance test for Figures 6, S15-S16 under the null hypothesis that the observed percentage (%) of agreement between compound events and CDEs is due to chance. To compute significance, we followed these steps: i) create n=1,000 datasets of random dates, with the same number of elements in each dataset as we have for the CDEs; ii) compute the % of agreement between compound events and the random dates for each iteration of the dataset and grid-point; iii) pool together all the random % values and compute the 1st and 99th quantiles for each grid-point; iv) check whether the observed % values fall outside these quantile values, and if this is the case consider the % values statistically significant at the 1% level (p-value <0.01). Since we obtain the vast majority of % as statistically significant, in the updated Figures 6, S15-S16 we show stippling for *non*-significant values. We described this statistical test in Section 2.3 of the revised paper and updated Figures 6, S15-S16 with new colorbars and stippling.

**C29:** L190, Do you think that re-computing the trends in the two metrics obtained based on maps of (1) land surface only and of (2) sea surface only could somehow allow for speculating more safely about this? Or, more in general, could this allow for disentangling a higher signal of the increasing coupling on land?

**A29:** Yes, computing the dynamical systems metrics based on land-surface only (and/or sea-surface only) data may help in providing an improved understanding of the physical processes at play during summer. Temperature-precipitation coupling may change significantly between land and sea, due to the very different thermal inertiae of the underlying surfaces, and the fact that in the former many components of the earth's surface affect the coupling (e.g. vegetation, orography, built environment and freshwater systems), whereas in the latter the Clausius-Clapeyron relation is followed with no (or little) disturbances. As suggested, we computed Figure 1 for land- and sea-only (Figure R_3) and found that JJA alpha trends are positive and significant for both land- and sea-only data, with the latter showing lower values compared to the former. The same trends are found for co-persistence over land, however co-persistence over the sea does not show statistical significance. We described these new findings in the revised paper (Section 3) and added Figure R_3 in the Supplementary Material.

[Figure]

*Figure R_3 - As Figure 1 but for ERA5 grid-points over (a)-(d) land- and (e)-(h) sea-only.*

**Remaining response to Referee 2**

**We thank the referee for taking the time to review our manuscript. Please, find below your remaining comment (CX) and our answer (AX), the latter highlighted in red.**

**C5:** 3) Please use colorbars with discrete colors for all map figures. For example, when I am interested in the SLP anomaly over Italy in Figure 4a it is very difficult to link the discrete colors on the map to the continuous colors in the colorbar.

**A5:** Thank you for your comment. We amended all the maps in the main text and Supplementary Material with discrete colorbars and also adjusted the colobars' limits to improve the comparison between the three reanalysis products (see also A6 of Referee 3).

[revised manuscript text omitted]

---

## Author Response (AR2)

**Response to Editor (2)**

**We thank the editor for taking the time to review our manuscript. Please, find below your comments and our answers, the latter highlighted in red. Lastly, you will find the revised manuscript with track changes highlighted.**

While going through the text again, I stumbled over one sentence, for which I would like to seek clarification before acceptance. In L 183 you write "We hypothesise that the large number of CDEs during July and August (Figures 3b and S7b) can be linked to extreme summertime precipitation events, that cool the air and increase wetness." Couldn't the high numbers of CDEs in summer also stem from warm-dry events? In particular, could the events in DJF and JA be of different type? E.g. in winter, we might have strong recurrence of cold-wet events and in summer we have strong recurrence of warm-dry events? Furthermore, couldn't this easily be checked by separate composite map for JA (similar to Fig. 4d,f but for JA)? If I'm misunderstanding something here, please clarify.

Thanks for pointing this out. As suggested, we computed Figure 4d,f but for July-August (JA) only (Figure R_1) and found warm-dry anomalies (instead of cold-wet). We amended the text accordingly in the manuscript (Section 4.1).

[Figure]

*Figure R_1 - As Figure 4d,f bur for July-August (JA) only.*

Also, isn't there a slight contradiction in the two sentences that follow: "We further note that, notwithstanding the previously mentioned correlation between co-persistence and alpha, the seasonality of θ−1 extremes – defined analogously to the CDEs – does not reflect that of the CDEs (not shown). For both variable combinations, the two shoulder seasons (i.e. spring and autumn) display very few CDEs." There is no correlation but the two shoulder seasons display very few CDEs in both cases?

In this case, the second sentence: "For both variable combinations, the two shoulder seasons (i.e. spring and autumn) display very few CDEs." refers to CDEs (i.e. alpha extremes) and *not* to θ−1 extremes. Therefore, for "both variable combinations" we meant CDEs computed from Tmax-P and Tmin-P. We made this clearer in the revised manuscript (Section 4.1).

[revised manuscript text omitted]